# LinEAS: End-to-end Learning of Activation Steering with a Distributional Loss

**Pau Rodríguez**[*]
Apple

**Michal Klein**
Apple

**Eleonora Gualdoni**
Apple

**Valentino Maiorca**
Sapienza, Apple

**Arno Blaas**
Apple

**Luca Zappella**
Apple

**Marco Cuturi**
Apple

**Xavier Suau**[*]
Apple

## Abstract

The growing use of generative models in daily life calls for efficient mechanisms to control their generation, to *e.g.,* produce safe content or provide users with tools to explore style changes. Ideally, such mechanisms should require low volume of unpaired data (*i.e.,* without explicit preference), and should be cheap, both at train and inference time, while preserving output quality. Recent research has shown that such mechanisms can be obtained by intervening exclusively on model *activations*, with the goal of correcting *distributional* differences between activations seen when using prompts from a source vs. a target set (*e.g.,* toxic and non-toxic sentences). While cheap, these fast methods are inherently crude: their maps are tuned locally, not accounting for their impact on downstream layers, resulting in interventions that cause unintended shifts when used out-of-sample. We propose in this work linear end-to-end activation steering (LinEAS), an approach trained with a global loss that accounts simultaneously for all layer-wise distributional shifts. In addition to being more robust, the loss used to train LinEAS can be regularized with sparsifying norms, which can automatically carry out neuron selection. LinEAS only requires a handful of unpaired samples to be effective, and beats similar baselines on toxicity mitigation in language models, becoming competitive with oracle-dependent methods that have access to strong supervision. LinEAS is modality-agnostic and we empirically find that it outperforms existing activation steering methods at mitigating and including new concepts at the output of single-step text-to-image generation models.

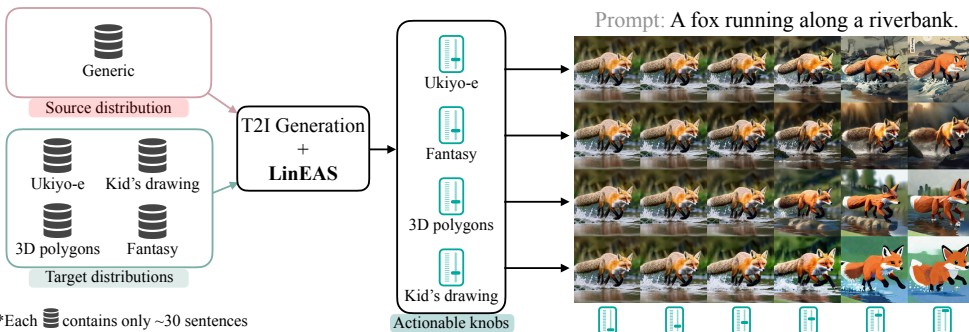

Figure 1: LinEAS learns lightweight maps to steer pretrained model activations. With LinEAS, we gain fine-grained control on text-to-image generation to induce precise styles (in the figure) or remove objects (*e.g.,* Section 4.4). The same procedure also allows controlling LLMs (*e.g.,* Section 4.1).

39th Conference on Neural Information Processing Systems (NeurIPS 2025).

# 1 Introduction

Modern generative models are typically trained in two distinct phases. The first phase, known as pre-training, involves learning from a large corpus of data using tasks like next-token prediction or text-to-image generation. This is followed by an alignment phase designed to adjust the model towards a more specific, desired behavior. This alignment can be achieved through instruction fine-tuning [1], reinforcement learning from human feedback (RLHF) [2] for LLMs, or guidance [3] and LoRA adapters [4] in text-to-image diffusion. Many of these approaches propose to modify the model's internal mechanisms, realigning its parameters by leveraging new data with, ideally, a minimal impact on the utility of the model.

The rapid growth of model sizes, coupled with the potentially infinite combination of alignment goals, calls for alignment mechanisms that readily adapt to new and evolving user needs. Ideally, adaptable alignment methods should adhere to the following desiderata: low training cost (potentially on device), memory-efficiency (small set of parameters), low inference time overhead, data-efficiency (few annotated samples), and fine-grained control. Working in low-data regimes makes data collection simple or even unmanned, and makes training potentially faster while fine-grained control makes alignment methods more customizable [5]. Such advantages together enable an agile control of generative models (see example in Figure 1), giving tools to users to customize their experience, and to administrators to quickly intervene to prevent harmful model behaviors.

A body of work known as *activation steering* [6–11] has proven to be effective at conditioning models while being memory and compute efficient, but still falls short in other desiderata. Some methods require data points to be *paired* with their corresponding counterfactuals [12], while others require a reward model (or a human) to indicate preference among generated outputs [2]. While this is a form of *strong supervision* that provides a robust signal to alignment methods, it is costly or even impossible to obtain in many scenarios. A weaker and more flexible form of supervision consists of dealing with two sets of examples – one for the target and one for the source behavior (*e.g.,* non-toxic and toxic sentences). In this setting, sentence pairs are not required, and no additional supervision is needed. Methods using this weaker signal typically analyze the distribution of activations from each set, thus we refer to them as a distributional approaches.

In this work, we propose Linear End-to-end Activation Steering (LinEAS), a **low-data** and **weakly-supervised** (unpaired data, no reward model) method that is trained with a **global distributional cost** grounded in optimal transport theory as signal for steering. Our hypothesis, which we validate empirically, is that our end-to-end learning accounts for the interactions between maps at different layers, leading to improved results over other steering methods. An additional advantage of optimizing a global cost is that it allows us to introduce additional objectives such as a regularization coefficient, which results in more targeted interventions, preserving the utility of the model.

Our main contributions are:

- We propose LinEAS, a novel framework to steer activations based on affine optimal transport maps between activations trained jointly across layers (*e.g.,* $T_1$, $T_2$ in Figure 2) with a global loss that enforces a global distributional alignment across all layers.[1] Our method provides low-budget conditioning, with a continuous and theory-grounded application strength $\in [0, 1]$ (*e.g.,* Figure 1).
- We show how LinEAS can be coupled with a *sparse lasso* regularizer [13] that can detect a small subset of activations that matter for a steering goal, reducing the intervened support by $100\times$, resulting in improved model utility.
- We show that LinEAS learns effective interventions with as few as 32 source and 32 target unpaired data points, showing state-of-the-art toxicity mitigation among steering methods.
- We validate fine-grained control of text-to-image (T2I) models, showing superior coherence with the prompt semantics while achieving the desired alignment goal.

# 2 Related Work

We classify steering methods based on whether they require strong (paired data or use of a human supervision / oracle) or weak supervision (unpaired data and no further supervision).

---

[1] `https://github.com/apple/ml-lineas`.

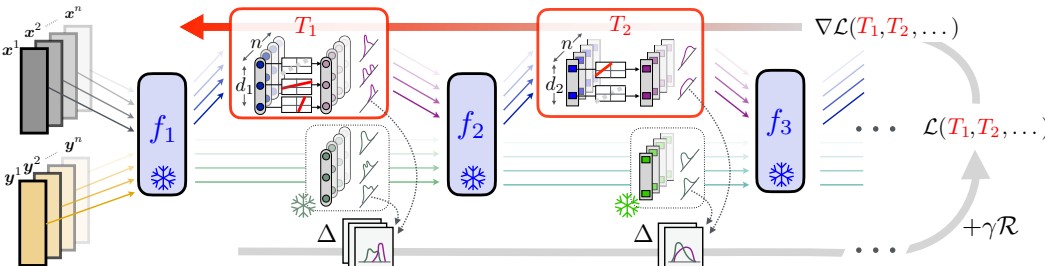

Figure 2: Given a frozen computational graph (blue) of $L + 1$ layers of interest, we interlace it with $L$ transport blocks (red). Each transport is defined as a collection of *coordinate-wise* affine transformations, as displayed in the 3 and 2 boxes for maps $T_1$ and $T_2$ respectively. All transport maps are jointly trained to minimize a sum of distributional losses $\Delta$ between the neural activation distributions collected from samples $\boldsymbol{x}_1, \ldots, \boldsymbol{x}_n \sim p$ (one shade of grey per sample) and $\boldsymbol{y}_1, \ldots, \boldsymbol{y}_n \sim q$ (resp. yellow). We learn the parameters of these maps *jointly* by minimizing the penalized sum of $\Delta$ terms, where $\Delta$ is a 1D Wasserstein distances evaluated on the $d_\ell$ activations of layer $\ell$. Using a global optimization, we can consider sparsifying regularizers (included in $\mathcal{R}$), to, *e.g.,* select a sparse subset of activations that require interventions. For instance, when adding a regularizer that promotes sparsity, both $T_1$ and $T_2$ do not intervene on one neuron, the first and the second, respectively.

**Strong supervision**. CAA [7] calculates the steering vector as the mean of differences between two sets of paired prompts. ReFT [14] optimizes low-rank projections of representations. LoFiT [11] learns a bias added to the representations of pre-selected attention heads. LoFiT training is similar to ReFT (although simpler, since only biases are trained). BitFit [15] directly finetunes the bias terms of pre-selected linear layers. Most of these methods can be viewed as a simplified version of LoRA [16]. Similarly to LoRA they also require either paired data or a human /oracle supervision during training.

**Weak supervision**. Some methods propose to add a vector to the activations. For example, ACTADD [17] uses the difference between 2 prompts and Mean-AcT the difference in means [10]. ITI-C [9] uses a steering vector orthogonal to the hyperplane learned by a binary linear classifier on the activations from two sets of sentences. With a different approach, AURA by Suau et al. [18] dampens activations proportionally to each neuron's ability to classify toxic and non-toxic sentences, effectively mitigating toxicity. REPE, by Zou et al. [8], does require paired data, however, we place this algorithm in the weak supervision family since it computes steering vectors based on a single prompt pair. Closest to our work, Lin-ACT [10] uses an affine map to steer activations. In Section 3.1 we discuss in more detail the differences between our proposed LinEAS with Lin-ACT and other methods.

## 3   End-to-end Learning of Steering Maps

We propose LinEAS, a method to optimize activation-specific interventions in a joint manner. Our hypothesis is that a global estimation that exploits causal interdependencies between activations across layers is needed to maintain the model's utility while achieving the steering goal.

### 3.1   Interventions Setup and Distributional Loss

We consider a generative model and target a set of $L$ intermediate activation layers. We describe the model as a composition of $L+1$ abstract pretrained functions, where the output, given an input prompt $\boldsymbol{x} \in \mathcal{S}$, can be written as $\boldsymbol{o} = f_{L+1} \circ f_L \circ \cdots \circ f_1(\boldsymbol{x})$. For convenience, we denote $d_\ell$ the dimension of the activations obtained at layer $\ell$, *i.e.,* the size of the activation vector $f_\ell \circ \cdots \circ f_1(\boldsymbol{x})$. We interlace this pretrained computational graph of $L + 1$ intermediate frozen layers with $L$ comparatively much simpler vector-to-vector maps: $\boldsymbol{o} = f_{L+1} \circ T_L \circ f_L \circ \cdots \circ T_2 \circ f_2 \circ T_1 \circ f_1(\boldsymbol{x})$, where for $1 \leq \ell \leq L$, the map $T_\ell : \mathbb{R}^{d_\ell} \to \mathbb{R}^{d_\ell}$ acts on the intermediate activations observed after layer $\ell$, by altering its coordinates and outputting a vector of the same size.

**Layerwise Distributional Losses.** We consider two distinct probability distributions over prompts, the source distribution $p$ and the desired target distribution $q$. For example, $p$ and $q$ could be such that a sample from the source distribution $p$ corresponds to toxic sentence, while a sample from the target

distribution $q$ corresponds to a non-toxic sentence (see Section 4.1). We then view each sampled prompt through the lens of their sequence of L activations. In practice, this means having access to samples $\boldsymbol{x}^1, \ldots, \boldsymbol{x}^n \sim p$ and $\boldsymbol{y}^1, \ldots, \boldsymbol{y}^n \sim q$, tracking their execution trace of their $\ell \leq L + 1$ activations, either modified through interleaved transports for samples from $p$:

$$\boldsymbol{\xi}_\ell^i := T_\ell \circ f_\ell \circ T_{\ell-1} \circ \cdots \circ T_1 \circ f_1(\boldsymbol{x}^i), \quad \ell \leq L, \tag{1}$$

or ran through the original network for samples of $q$:

$$\boldsymbol{\eta}_\ell^j := f_\ell \circ \cdots \circ f_1(\boldsymbol{y}^j), \quad \ell \leq L. \tag{2}$$

Our goal is to learn *jointly* all $L$ transport maps so that, for each $\ell \leq L$, the families of vectors $(\boldsymbol{\xi}_\ell^i)_i$ and $(\boldsymbol{\eta}_\ell^j)_j$ are *similar* with respect to a distributional metric $\Delta$ between probability measures, making the cost below small:

$$\mathcal{C}(T_1, \ldots, T_\ell; (\boldsymbol{x}^i)_i, (\boldsymbol{y}^j)_j) = \sum_{\ell \leq L} \Delta((\boldsymbol{\xi}_\ell^i)_i, (\boldsymbol{\eta}_\ell^j)_j). \tag{3}$$

**Sliced Wasserstein Losses.** To define $\Delta$ at each layer, we adopt the approach of Rodriguez et al. [10] and sum $d_\ell$ univariate Wasserstein distances between the $d_\ell$ marginal distributions at each layer $\ell$. This choice is motivated by the fact that in the typical setting targeted in this work, we must deal with a high-dimensionality / low sample regime, $d_\ell \gg N$, that would hinder the use of more complex multivariate distributional losses that account more closely for cross-variable effects. We observe that adding univariate quantities yields a more robust loss estimation that translates to better downstream tasks than considering, e.g., Sinkhorn divergences [19].

To define $\Delta$ at each layer, we adopt the approach of Rodriguez et al. [10] and use $d$ univariate Wasserstein distances between the $d$ marginal distributions. The activations can be arranged as matrices $U := [\boldsymbol{\xi}_\ell^1, \ldots, \boldsymbol{\xi}_\ell^n]$ and $V = [\boldsymbol{\eta}_\ell^1, \ldots, \boldsymbol{\eta}_\ell^n]$, both in $\mathbb{R}^{n \times d_l}$. To compute their 1D-Wasserstein distance [20, Chap. 2], these activations must be first sorted in increasing order along the feature axis:

$$\tilde{U} = \text{sort}(U, \text{axis} = -1), \tilde{V} = \text{sort}(V, \text{axis} = -1)$$

to define the sliced Wasserstein distance [21] computed only on the *canonical directions*, namely:

$$\Delta(U, V) := \sum_{j=1}^d W_2^2(U_{\cdot j}, V_{\cdot j}) = \frac{1}{n} \sum_{j=1}^d \|\tilde{U}_{\cdot j} - \tilde{V}_{\cdot j}\|^2. \tag{4}$$

**Differences with ITI-C, Lin-ACT and ReFT.** Both ITI-C [9] and Lin-ACT [10] optimize each $T_\ell$ *independently* across layers, minimizing a single distributional difference in closed form (Lin-ACT) or learning a linear classifier (ITI-C), and assuming all other layers are frozen. While arguably much faster, this also generates causal inconsistencies, which can be partially resolved as in Lin-ACT by using a sequential approach: when $T_{\ell-1}$ is trained, $T_{\ell-1}$ is reused to recompute all activation distributions used for $T_\ell$. We claim that this suboptimality is to blame for poor generalization. This independent approach also precludes trade-offs when choosing which activations to turn on/off across layers, which we can easily be surfaced using sparsity regularizers. One fundamental difference with ReFT [12] is that LinEAS does not need paired data, *i.e.,* sample $\boldsymbol{y}^i$ does not need to be a counterfactual of $\boldsymbol{x}^i$, which is the case for ReFT. While counterfactual data is important for applications like translation, there is a vast amount of applications (*e.g.,* toxicity) where paired data is not available.

## 3.2 Parameterization and Regularization

Building on the approach outlined by Rodriguez et al. [10], we propose to parameterize each map $T_\ell$ as an affine map for each activation $\ell \leq L$, namely for $z \in \mathbb{R}^{d_\ell}$, one has

$$T_\ell(z) := \omega_\ell \odot z + b_\ell, \quad \omega_\ell, b_\ell \in \mathbb{R}^{d_\ell}, \tag{5}$$

where $\odot$ is the element-wise product. We write $\mathbf{w} := (\omega_1, \ldots, \omega_L)$ and $\mathbf{b} := (b_1, \ldots, b_L)$ for the collections of all scale and intercept parameters. In what follows since each map $T_\ell$ is entirely parameterized through its scale/intercept parameters $\mathbf{w}, \mathbf{b}$, we overload notations to define

$$\mathcal{C}(\mathbf{w}, \mathbf{b}; (\boldsymbol{x}^i)_i, (\boldsymbol{y}^j)_j) := \mathcal{C}(T_1, \ldots, T_{\ell-1}; (\boldsymbol{x}^i)_i, (\boldsymbol{y}^j)_j).$$

**Sparse Regularization.** We propose to use a sparsity regularizer that will carry out both *layer* and *within-layer* selection of activations. This can be achieved by using structured regularization, using either 1-norms or 2-norms:

$$\mathcal{R}_1(\mathbf{w}, \mathbf{b}) := \sum_\ell \|\omega_\ell - \mathbf{1}\|_1 + \|b_\ell\|_1 \quad \text{and} \quad \mathcal{R}_G(\mathbf{w}, \mathbf{b}) := \sum_\ell \sqrt{d_\ell} \left(\|\omega_\ell - \mathbf{1}\|_2 + \|b_\ell\|_2\right).$$

resulting in a *sparse group lasso* regularizer [13, 22], $\mathcal{R} := \lambda_1 \mathcal{R}_1 + \lambda_G \mathcal{R}_G$, which can be added to the cost to result in:

$$\mathcal{L}(\mathbf{w}, \mathbf{b}) := \mathbb{E}_{\substack{(\boldsymbol{x}^i)_i \sim p, \\ (\boldsymbol{y}^j)_j \sim q}} \left[\mathcal{C}(\mathbf{w}, \mathbf{b}; (\boldsymbol{x}^i)_i, (\boldsymbol{y}^j)_j)\right] + \gamma \mathcal{R}(\mathbf{w}, \mathbf{b}), \tag{6}$$

where $\gamma$ controls the amount of sparsity, *i.e.,* larger $\gamma$ will result in fewer activations and/or layers being intervened on, with solution such that $\omega \approx \mathbf{1}$ and $b \approx \mathbf{0}$.

**On the choice of sparsity hyperparameters** We only consider two hyperparameters at the moment, $\lambda_1$ and $\lambda_G$, which is relatively small since LinEAS intervention models have none. Empirically, we found that tuning $\gamma \in [0, 1]$ with $\lambda_1 = \lambda_G = 1$ already provides interesting trade-offs between conditioning and utility as reported in Figures 4 and 8. This also reduces the number of additional hyper-parameters to tune to just one ($\gamma$) and enables automatic layer selection (Figure 10).

### 3.3 Optimization

**Proximal SGD.** We optimize $\mathcal{L}$ in (6) with proximal stochastic gradient descent (PSGD) with a learning rate of 0.1 and cosine decay. We assume access to 2 sets of unpaired $N$ prompts $(\boldsymbol{x}^i)_{i=1}^N$ and $(\boldsymbol{y}^i)_{i=1}^N$ and run PSGD on minibatches of activations $(\boldsymbol{\xi}_\ell^i)_i$ and $(\boldsymbol{\eta}_\ell^i)_i$ of size $n$. Note that activations originating from $\boldsymbol{y}^i$ use the untouched network, and can be pre-computed beforehand. In Appendix D we provide details on the algorithm and proximal operators.

## 4 Experimental Results

### 4.1 Toxicity Mitigation in LLMs

We analyze the effectiveness of LinEAS at the important task of toxicity mitigation. To that end, we compare with prompting, CAA [7], ReFT [12], ITI-C [9] and Lin-ACT [10] on three LLMs ranging from 1.5B to 7B parameters, by aligning the activations of $N = 32$ toxic to 32 non-toxic sentences sampled from the Jigsaw dataset [23].

**Toxicity Metrics.** We evaluate toxicity mitigation on the *RealToxicityPrompts* (RTP) dataset [24] and the *Thoroughly Engineered Toxicity* (TET) dataset [25]. For RTP, we follow Rodriguez et al. [10] by sampling 1000 prompts from the dataset and let the model (intervened or not) complete them. For TET, we use the 2546 prompts provided. Then, we score the generations using the open-source Roberta toxicity classifier (RTC) [26]. We report $\text{Tox}_{\text{RTC}}^{\text{RTP}}$ and $\text{Tox}_{\text{RTC}}^{\text{TET}}$, the respective percentage of generations flagged as toxic on RTP and TET.

**Utility Metrics.** To measure whether the utility of the model is affected by these interventions, we report $\text{PPL}_{\text{WIK}}$, the perplexity obtained on a fixed set of 20k Wikipedia sentences [27], as well as the overall 5-shot accuracy on the MMLU compendium [28].

**Oracle Baselines.** In addition, we introduce two baselines that require a strong supervision signal directly from the RTC oracle, *i.e.,* the classifier used to compute *test-time metrics*, giving them a significant advantage. More precisely, we train a LoRA adapter and LoFIT [11] using the RTC labels, yielding the oracle methods LRTC-RL and LoFIT-RL. We report in Table 1 only the results of LoFIT-RL as it performed better than LRTC-RL (available in Table 7). See Appendix I for details on the oracle training protocol.

**Setup.** All methods have access to only 32 toxic and 32 non-toxic sentences (unpaired). We optimize LinEAS for 1K steps using SGD and a learning rate of 0.1. As we focus on the benefit of using an end-to-end loss, we use $\gamma = 0$ for LinEAS (see Section 4.3 for $\gamma$'s impact). For Lin-ACT and CAA, we use their default settings, and set intervention strength to $\lambda = 1$. For ReFT, we train for 10 epochs (selected with an epoch sweep). For ITI-C we use $\lambda = 0.5$, obtained through grid search.

We evaluate toxicity mitigation by intervening upon different layers, and report the best overall layer type per method, namely `.*post.*layernorm` for LinEAS and `.*o_proj` for ITI-C and Lin-AcT, according to the Huggingface implementation of the models. Both CAA and ReFT intervene upon the residuals of the Transformer blocks, as suggested in their original works. For CAA we run two baselines, intervening on all layers (reported in Table 1), and intervening on the middle layer of each model (as in the original paper, reported in Table 7 given its worse performance). Additionally, we include prompting as a conditioning strategy, where the model is preprompted with *"Continue the text in a non-toxic way:"*.

| Model | Method | #Params | $\text{Tox}_{\text{RTC}}^{\text{RTP}}$ ($\downarrow$) | $\text{Tox}_{\text{RTC}}^{\text{TET}}$ ($\downarrow$) | $\text{PPL}_{\text{WIK}}$ ($\downarrow$) | MMLU ($\uparrow$) |
|---|---|---|---|---|---|---|
| | None | - | $3.00_{0.54}$ | $23.09_{0.67}$ | $13.67_{0.00}$ | $60.95_{0.00}$ |
| | LoFIT-RL | 0.86M | $0.37_{0.06}$ | $4.36_{0.00}$ | $14.12_{0.07}$ | $59.74_{0.14}$ |
| | Prompt | - | $4.07_{0.38}$ | $21.02_{1.44}$ | $13.65_{0.00}$ | $60.96_{0.00}$ |
| Qwen2.5-1.5B | CAA⋆ | 0.043M | $1.15_{0.37}$ | $5.77_{2.14}$ | ${\color{red}19.30}_{2.76}$ | ${\color{red}37.67}_{6.95}$ |
| | ReFT⋆ | 0.39M | $2.57_{0.60}$ | $18.17_{3.04}$ | $15.58_{0.52}$ | $58.84_{0.23}$ |
| | ITI-C | 0.043M | $1.87_{0.21}$ | $18.16_{0.62}$ | $12.39_{0.09}$ | $60.88_{0.08}$ |
| | Lin-AcT | 0.086M | $1.50_{0.35}$ | $13.88_{1.72}$ | $13.88_{0.16}$ | $60.09_{0.25}$ |
| | LinEAS | 0.086M | $\mathbf{1.07}_{0.46}$ | $\mathbf{12.70}_{0.74}$ | $14.10_{0.07}$ | $59.97_{0.16}$ |
| | None | - | $4.00_{0.45}$ | $13.39_{1.42}$ | $14.79_{0.00}$ | $53.03_{0.00}$ |
| | LoFIT-RL | 0.11M | $0.40_{0.20}$ | $1.76_{0.00}$ | $15.43_{0.08}$ | $52.17_{0.17}$ |
| | Prompt | - | $4.60_{0.36}$ | $12.32_{0.67}$ | $14.81_{0.00}$ | $53.18_{0.00}$ |
| Gemma2-2B | CAA⋆ | 0.06M | $0.80_{0.00}$ | $2.44_{1.99}$ | ${\color{red}23.52}_{2.67}$ | ${\color{red}26.86}_{0.08}$ |
| | ReFT⋆ | 0.54M | $2.85_{0.49}$ | $11.15_{1.91}$ | ${\color{red}19.93}_{0.30}$ | $48.99_{1.34}$ |
| | ITI-C | 0.06M | $1.17_{0.60}$ | $7.15_{0.92}$ | $14.00_{0.11}$ | $52.78_{0.23}$ |
| | Lin-AcT | 0.12M | $1.60_{0.32}$ | $7.76_{0.39}$ | $14.78_{0.12}$ | $52.43_{0.57}$ |
| | LinEAS | 0.24M | $\mathbf{0.73}_{0.10}$ | $\mathbf{4.02}_{0.68}$ | $15.46_{0.21}$ | $52.22_{0.40}$ |
| | None | - | $3.92_{0.59}$ | $25.16_{0.92}$ | $10.67_{0.00}$ | $74.26_{0.00}$ |
| | LoFIT-RL | 0.10M | $1.10_{0.38}$ | $7.11_{0.30}$ | $10.91_{0.16}$ | $73.87_{0.17}$ |
| | Prompt | - | $6.80_{0.00}$ | $21.22_{0.21}$ | $10.65_{0.00}$ | $74.23_{0.00}$ |
| Qwen2.5-7B | CAA⋆ | 0.10M | $1.20_{0.00}$ | $9.25_{3.07}$ | $12.83_{0.00}$ | ${\color{red}48.58}_{0.00}$ |
| | ReFT⋆ | 0.90M | $3.33_{0.96}$ | $20.38_{2.37}$ | $13.80_{1.20}$ | $70.43_{0.60}$ |
| | ITI-C | 0.10M | $2.63_{0.44}$ | $19.98_{1.24}$ | $9.63_{0.03}$ | $74.08_{0.05}$ |
| | Lin-AcT | 0.20M | $2.72_{0.46}$ | $21.64_{2.00}$ | $11.42_{0.34}$ | $72.18_{0.16}$ |
| | LinEAS | 0.20M | $\mathbf{1.95}_{0.48}$ | $\mathbf{14.95}_{0.92}$ | $10.91_{0.35}$ | $73.67_{0.05}$ |

Table 1: Toxicity mitigation on the RTP and TET datasets using three different models, Qwen2.5-1.5B, Gemma2-2B, and Qwen2.5-7B. Strongly degraded utility is marked in ${\color{red}\text{red}}$. We report results at low data regime ($N = 32$ sentences to estimate the interventions). See Appendix E for more models, baselines and an ablation with larger training size and different intervention layers. Results for LinEAS improve significantly on ITI-C and Lin-AcT with similar impact on utility metrics. The quality of these interventions is often on par with the oracle baseline LoFIT-RL in terms of utility, although the strong oracle supervision yields better mitigation. ⋆*CAA and ReFT are designed to use paired data, which is not available in the toxicity setting.*

**Results.** Table 1 summarizes the toxicity mitigation experiments averaged over 4 generation seeds (and RTP samplings). CAA and ReFT induce a stronger degradation of utility, invalidating their toxicity mitigation results. Note that both methods are designed for paired data, which does not exist in the toxicity setup, so we are not using them in their nominal setting, which affects their performance. Prompting is not effective for the models tested, and even increases $\text{Tox}_{\text{RTC}}^{\text{RTP}}$. LinEAS achieves a consistent toxicity mitigation, outperforming all steering methods at this low data regime. For example, LinEAS reduces Gemma2-2B toxicity by $5.5\times$, getting closer to the mitigation obtained with the oracle LoFIT-RL. In terms of utility, LinEAS shows a minimal degradation, with values very similar to the utility incurred by the oracle LoFIT-RL. In absolute terms, LinEAS reduces MMLU by less than 1 point and increases $\text{PPL}_{\text{WIK}}$ by less than 0.6. In Table 9 (Appendix E) we show that LinEAS is much more robust to the choice of layer than ITI-C and Lin-AcT. Additionally, we also provide in Figure 3 (and Table 8) an analysis on a higher data regimes, up to $N = 1024$ sentences, showing that LinEAS is also reliable in such scenario. Beyond the differences in parameter sizes between ITI-C, Lin-AcT and LinEAS on the one hand, and the LRTC-RL approach on the other, we also note that the compute needed to train these methods is significantly different: in the low data regime $N = 32$, estimating each method on a Nvidia A100-80Gb GPU and Qwen2.5-7B takes 37s (ITI-C), 30s (Lin-AcT) leveraging closed forms, 500s (LinEAS for 1K steps) and 27300s for LoFIT-RL, see Appendix B for a deeper analysis.

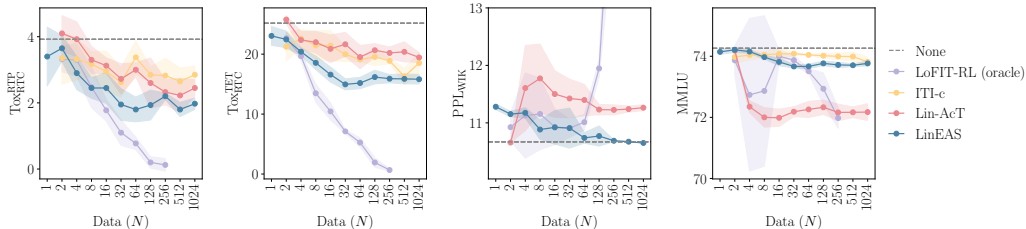

Figure 3: **LinEAS is effective at low data regime.** We study toxicity mitigation (two left-most plots) and utility (two right-most plots) as a function of the amount of data available to learn interventions. LinEAS shows better performance (low toxicity and utility close to original dashed lines) for low data, and stable performance for $N \geq 32$.

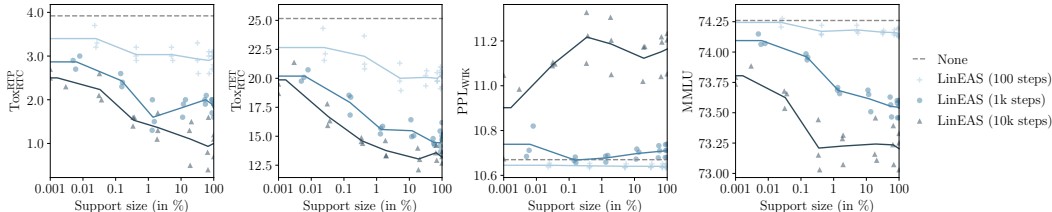

Figure 4: **Sparsity improves utility while mitigating toxicity**. Toxicity results on Qwen2.5-7B using only 32 sentences, at different levels of sparsity $\gamma$ that result in different support sizes (x axis). At 1K optimization steps, with a support of about 1% we maintain similar toxicity (left, center-left) while $PPL_{WIK}$ decreases (center-right) and MMLU increases (right). Note that too long optimizations (10k steps) might harm utility, due to overfitting. Similarly, short optimizations (*e.g.,* 100 steps) and strong sparsity leads to low conditioning (mild toxicity mitigation).

**User Study.** We complement the quantitative results with a user preference study to evaluate the perceived quality of continuations generated by different intervention methods. Our findings indicate a strong preference for LinEAS over three other alternatives: Lin-AcT, ITI-c, and no intervention (identity). Specifically, users preferred LinEAS in $57.70\%$ of cases compared to $18.43\%$ for Lin-AcT and $11.67\%$ for ITI-c. We provide more details in Appendix C.

## 4.2 Effect of Data on Toxicity Mitigation

With the same toxicity setting as in Section 4.1, we ablate the amount of data used to estimate interventions using Qwen2.5-7B (best model studied in terms of MMLU). We sweep $N$ from 1 to 1024, meaning we have access to $N$ toxic and $N$ non-toxic sentences. Note that the results in Table 1 correspond to $N = 32$. In Figure 3 we plot the evolution of toxicity ($Tox_{RTC}^{RTP}$ and $Tox_{RTC}^{TET}$) and utility ($PPL_{WIK}$ and MMLU) as a function of $N$, averaged over 4 random sweeps (standard deviation as shaded areas). LinEAS achieves superior toxicity mitigation even at very low data regimes, while maintaining utility close to the original model (horizontal dashed lines) and the LoFIT-RL oracle. Moreover, LinEAS's performance is more stable for a large range of $N$ (32 to 1024), even more stable than the oracle which diverges for $N > 128$. Note that we fix the hyper-parameters for all the methods, including the oracle, which shows to be more sensitive to the training setting.

## 4.3 Effect of Sparsity on Toxicity Mitigation

Intuitively one should only steer the smallest set of activations needed to achieve a desired goal in order to preserve utility and keep most of the model's inference graph untouched. In this section, we explore how sparsity affects toxicity mitigation in the setup of Section 4.1 as we increase $\gamma$ from 0 to 0.1. Increasing $\gamma$ results in less activations being *transported*, which we measure as support $= \|(\mathbf{w} \neq \mathbf{1}) + (\mathbf{b} \neq \mathbf{0})\|_0$, *i.e.,* all activations transported either by rescaling or shifts.

In Figure 4 we show how $Tox_{RTC}^{RTP}$ and $Tox_{RTC}^{TET}$, as well as the utility $PPL_{WIK}$ and MMLU, evolve as the sparse support decreases (x axis), for Qwen2.5-7B in the low ($N = 32$) data regime. We show the results of 3 sweeps of $\gamma$ with different random seeds (markers), and plot the average at

each $\gamma$ level (line). Note that at $\gamma = 0$ the support is approximately 100%. In the case of 1K steps (reported in Table 1), one can afford reducing the support to about 1% and still maintain the toxicity mitigation values at 100% support. Interestingly, at these support values, the $\text{PPL}_{\text{WIK}}$ and MMLU improve, validating our hypothesis that smaller supports help preserve the utility. We also observe that short optimizations (*e.g.,* 100 steps) lead to mild conditioning (poor toxicity mitigation) while long optimizations (*e.g.,* 10k steps) lead to a gradual degradation in utility. In Figure 8 (Appendix G) we show the same plot for the high data regime, with similar conclusions. Additionally, in Appendix L we study how the similarity of LinEAS interventions is correlated with human judgment, on pairs of concepts from the MEN dataset [29], showing strong correlation when using sparsity. We provide a similar analysis for T2I generation in Appendix H.

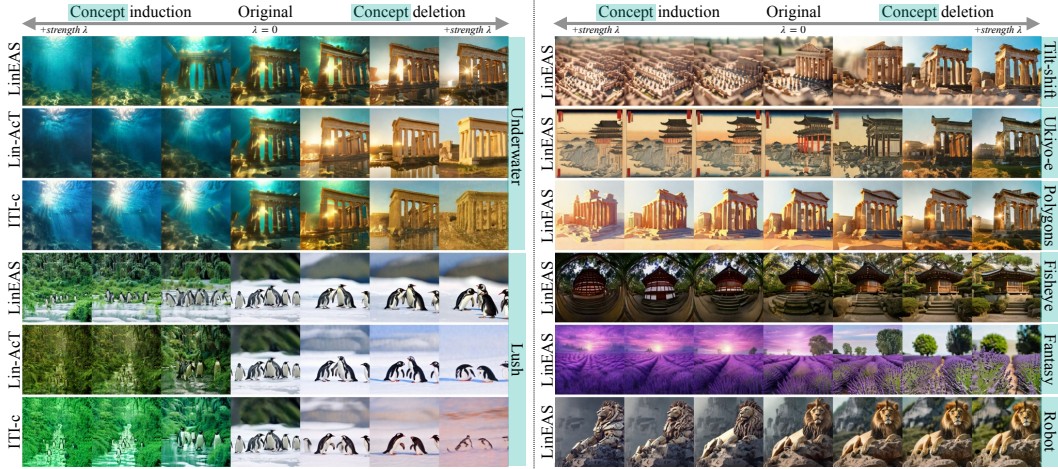

Figure 5: Generations using DMD2 [30]. (Left) Concept deletion using ITI-C, Lin-AcT and LinEAS for two different concepts, starting from prompts that contain the concept. LinEAS shows a more gradual deletion ($\lambda = 0.4, 0.7, 1$), and better preservation of the original image ($\lambda = 0$). (Right) Qualitative examples of LinEAS on 6 more concepts. We also show that inverting the steering maps surprisingly results in concept induction, probably due to strong structure in activation space. LinEAS also outperforms the other methods under this setting.

## 4.4 Steering Text-to-Image Generation Models

In this section, we explore how LinEAS can be used to remove the presence of concepts in text-to-image (T2I) generation: a task that plays a key role in generation alignment, similar to toxicity mitigation in LLMs. We preserve the focus on low data and compute budget, so we apply LinEAS to DMD2 [30], a recent single-step text-to-image generation model distilled from SDXL [31] with a GAN [32] loss. We note that there are diffusion guidance methods available to condition these models; however, they typically require multiple denoising steps (even up to 200) [33, 34] and, at times, trajectory resampling and evaluation strategies [35]. These additional overheads make them suitable for settings with higher compute budget, which differs from the low-compute setting we are interested in.

**Setup.** As in [10], we modulate the strength of LinEAS applied to all `layernorm` layers by introducing a scale $0 \leq \lambda \leq 1$ when applying interventions. Intuitively $\lambda = 0$ results in no intervention, while $\lambda = 1$ carries out a full LinEAS transport, any value in between reflecting a gradual change. Following Section 4.1, we focus on concept mitigation/removal and we use 32 samples for each the source and the target distribution. We train LinEAS for 1000 steps with batch size 4, AdamW, learning rate of $1e^{-4}$ and $\gamma = 0$. Find additional implementation details in Appendix N. We compare LinEAS with ITI-C and Lin-AcT, also weakly supervised methods that work with *unpaired data*.

**Data.** We query an open-source LLM for a diverse set of prompts covering 3 different conditioning categories and 5 different concepts per category with 32 prompts per concept, totalling 480 prompts. **Styles**: `vaporwave, lush, low-poly, ukiyo-e, fantasy`; **objects**: `robots, axolotl, book, car, hourglass`; and **perspective**: `macro, fisheye, bokeh,`

| Method | ↑ User Pref. | ↑ IMGSc. | ↓ CLIPSc. |
|---|---|---|---|
| ITI-C | $12.4_{5.5}\%$ | $0.24_{0.19}$ | $0.19_{0.02}$ |
| Lin-AcT | $24.4_{7.0}\%$ | $0.45_{0.21}$ | $0.18_{0.03}$ |
| LinEAS | $\mathbf{63.3}_{6.6}\%$ | $\mathbf{0.66}_{0.19}$ | $0.18_{0.03}$ |

Table 2: LinEAS mitigates concepts on DMD2 [30] while staying perceptually similar to the original image. Users prefer LinEAS 63.3% of the time (left) since it maintains a higher fidelity to the non-intervened original model when using the same prompt (center), and matches other methods at concept removal (right). Results were obtained with $\lambda = 1$ and they are aggregated across all concepts.

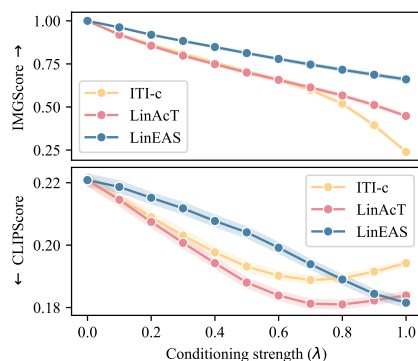

Figure 6: IMGScore and CLIPScore at multiple conditioning strengths ($\lambda$). (top) LinEAS is more faithful to DMD2 with $\lambda = 0$. (bottom) LinEAS is more consistent and predictable at mitigation.

`underwater`, and `tilt-shift`. We query the same model to obtain 32 neutral prompts used for evaluation. We provide a complete description and the prompts themselves in Appendix O.

**Metrics.** We measure (1) **CLIPScore** [36], the cosine similarity between CLIP embeddings of the generated images and the concept description. (2) **IMGScore**, the cosine similarity between DINOv2-small [37] image embeddings generated with $\lambda > 0$ and images generated with $\lambda = 0$ using the same prompts. CLIPScore assesses whether the generations are truthful to the desired style, and IMGScore whether they are perceptually similar to those generated without intervention.

**User study.** Aiming at complementing the quantitative analysis, we also run a user study. We consider 20 prompts × 5 conditioning concepts, yielding a total of 100 pairs. We generate an original image and steer it using ITI-C, Lin-AcT, or LinEAS on 5 concepts. We ask a pool of 10 participants to select their preferred output (a total of 1000 evaluations), showing a strong preference for LinEAS, as reported in Table 2 (left). The actual question asked to the participants is *"Which image blends best prompt and style, while remaining faithful to the untouched output?"*.

**Results.** Table 2 summarizes the results for the text to image evaluation. We found that 63% of the users prefer LinEAS (24.4% for Lin-AcT, and 12.4% for ITI-C). These results are in agreement with the automated metrics: LinEAS is *significantly* more faithful to the images produced by DMD2 with $\lambda = 0$ for the same prompts, with an IMGScore of 0.66 compared to 0.45 for Lin-AcT and 0.24 for ITI-C while all methods achieve a similar CLIPScore. We report more granular per-concept scores in Appendix N.2. Figure 6 explores how IMGScore and CLIPScore change with $\lambda$. We find that LinEAS is consistently more faithful to DMD2 with $\lambda = 0$ in terms of IMGScore (top) while showing a strong linear correlation with CLIPScore, making LinEAS more consistent and predictable.

**ⵔⵏⴹEⴷⵙ.** Surprisingly, inverting the affine maps in LinEAS: $T_\ell^{-1}(z) := (z - b_\ell) \odot \frac{1}{\omega_\ell}$ tends to negate the conditioning thus switching from mitigation to induction and vice-versa (see "concept induction" in Figure 5). We speculate that this behavior is due to a strong structure in the activation space. Quantitative results using the inverse maps can be found in Appendix N.2.

### 4.5 Layer Selection Analysis

To evaluate the robustness of LinEAS with respect to the choice of intervened layers, we conducted a sweep over different layer types within the DMD2 UNet. The results, averaged over 15 concepts, are presented in Table 3. We measure image consistency using IMGScore (where higher is better) and concept removal using CLIPScore (where lower is better).

Our findings indicate that while intervening on all layer normalization modules — the setting used in our main experiments — provides the best trade-off between the two metrics, LinEAS demonstrates robust performance across all tested layer configurations. This suggests that activation steering is not overly sensitive to the specific layer choice in UNets.

| Intervened Layers | # Modules | IMGScore ↑ | CLIPScore ↓ |
|---|---|---|---|
| All Normalizations | 256 | $0.714 \pm 0.054$ | $0.131 \pm 0.033$ |
| Transformer MLPs | 70 | $0.780 \pm 0.053$ | $0.137 \pm 0.031$ |
| All attention K and Q projections | 280 | $0.935 \pm 0.016$ | $0.157 \pm 0.028$ |
| All attention V projections | 140 | $0.818 \pm 0.051$ | $0.140 \pm 0.031$ |
| All attention input projections | 11 | $0.923 \pm 0.017$ | $0.155 \pm 0.028$ |
| Resnet Normalizations | 34 | $0.896 \pm 0.023$ | $0.156 \pm 0.027$ |

Table 3: Study on the choice of intervened layers for LinEAS in the DMD2 UNet. We report image consistency (IMGScore ↑) and concept removal (CLIPScore ↓), averaged over 15 concepts. The method shows robustness, with the default setting (all layer norms) offering the best trade-off.

## 5   Limitations and Open Problems

While the field of activation steering has gained considerable momentum there are some limitations that affect practically every method, including ours.

**Compositionality.** One such limitation is the ability to compose multiple interventions (learnt separately) so that multiple steering objectives are satisfied at the same time. We refer to this ability as *compositionality*, and we argue its difficulty lies in the fact that multiple interventions can interfere with one another and produce unexpected results. Our initial hypothesis was that sparsity could help mitigate such unwanted interference, since interventions for different concepts would act on (almost) disjoint sets of neurons. In Appendix J we present an analysis where we intervene for two concepts simultaneously, with different sparsity $\gamma$s. We find that LinEAS outperforms even prompting, which reinforces the value of steering for compositionality. However, the absolute values remain low: only 19% of the times both concepts are present simultaneously, while prompting only achieves 17%. We observe that most of the gain comes from the end-to-end optimization, which reaches 16% probability without sparsity, which is a $15\times$ improvement over Lin-AcT (no end-to-end optimization). Adding group lasso regularization improves results of LinEAS by an additional 3%. These results show, on the one hand that compositionality is still a challenging task, and on the other hand that there is room to investigate more suitable sparse regularizers in order to overcome the current limitations.

**Intervention selectivity.** Another common limitation of steering mechanisms that also affects LinEAS the intervention is applied to all tokens, usually with the goal of keeping inference cost in a budget. Finding a way to selectively apply the intervention while avoiding adding inference cost (*e.g.,* avoid using a classifier to decide on which tokens to intervene or not) remains an open problem.

## 6   Conclusion

We propose LinEAS, a novel framework to learn lightweight interventions on activations to steer model generation towards a desired property. LinEAS achieves state-of-the-art performance among steering methods on safety applications, such as avoiding toxic outputs, or style changes, both for LLMs (Gemma2-2B, Qwen2.5-1.5B and Qwen2.5-7B) and text-to-image generation (DMD2). Our approach learns a set of univariate maps that reshape a source to a target activation distribution, with an improved loss that yields improved controllability and robustness. Unlike previous methods, such as [10, 9], which require local adjustments and manual layer selection, our method optimizes the transformation globally in an end-to-end fashion. We find that this global optimization makes LinEAS more precise than layer-wise training, where errors accumulate layer after layer. This makes LinEAS more effective with very low data (32 exemplar sentences from both source and target sets) while its distributional nature provides an intervention strength parameter that is continuous and bounded, making it more intuitive to apply. This approach also allows for the incorporation of flexible regularizers, such as group-sparse and sparse, allowing for automatic selection of layers and/or neurons, leading to improved utility.

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

# A  Broader Impact

This paper presents an algorithm that aims to advance the field of Machine Learning without a specific application in mind.

Our objective has been to use our algorithm to condition models towards desired behaviors (*e.g.,* being less toxic), however, a malicious user with access to the model's activations could condition the model to behave in a negative way, *e.g.,* forcing the model to be more toxic.

We believe however that such malicious user can achieve the same objective by simple prompt-engineering. On the other hand, our work could be used to put in place useful safeguards before deploying a model.

# B  Hardware and Compute Requirements

The experiments in this work were computed on a single NVIDIA A100 GPU with 80GB RAM and they could also fit in an NVIDIA A100 with 40GB RAM.

**Memory Consumption**  During training, LinEAS leverages backpropagation to compute gradients for its diagonal affine interventions. This design ensures that only activations relevant to the intervened layers require storage during the forward pass, leading to a substantially reduced memory footprint compared to full parameter tuning.

**Compute**  Compared to local methods, the computational cost of LinEAS is primarily determined by the number of optimization steps required for intervention training. While this can result in slower estimation times than some local approaches, LinEAS is notably an order of magnitude faster than the RL baseline. Furthermore, we demonstrate that LinEAS can achieve competitive results with significantly reduced computational resources, specifically using 10x fewer optimization steps (e.g., 100 steps, approximately 50s for estimation) than the full configuration presented in the paper (Figure 4). This contrasts favorably with Lin-AcT's ~30s and dramatically outperforms LoFIT-RL's 27300s (Table 4). Overall, LinEAS offers a positive trade-off between computational expenditure and conditioning performance.

**Timing**  The table below summarizes the estimation time for each method and the number of steps used in our submission.

Table 4: Estimation times for various methods and models.

| Method | # Steps | Gemma2 (s) | Qwen1.5B (s) | Qwen7B (s) |
|---|---|---|---|---|
| LoFIT-RL | 100 | 7600 | 25900 | 27300 |
| ReFT | 10 | 92 | 80 | 100 |
| ITI | 1 | 29 | 30 | 37 |
| Lin-AcT | 1 | 17 | 14 | 22 |
| LinEAS | 1000 | 430 | 340 | 500 |

We have also timed DMD2 for T2I generation. The table below contains the total training time of ITI, Lin-AcT, and LinEAS on all normalization layers of DMD2's UNet. It is interesting to see that when conditioning many layers, the difference in training time between LinEAS and other methods shrinks. This is because LinEAS leverages PyTorch's backpropagation, which is optimized compared to ITI and Lin-Act's layer-wise estimation methods.

Table 5: Total training time for methods on DMD2's UNet.

| Method | # steps | DMD2 |
|---|---|---|
| ITI | 1 | 26m 44s |
| LinAcT | 1 | 25m 53s |
| LinEAS | 1000 | 29m 27s |

### B.1 Detailed Complexity Analysis

**Variables:**

- $B$: batch size
- $B \log B$: sorting cost for optimal transport
- $F$: cost of forward (backward) pass
- $N$: Number of SGD steps
- $T$: Number of logistic regression L-BFGS steps
- $I$: Number of intervened layers

**Computational Costs:**

- The computational cost of LinEAS is dominated by $O(N \cdot (2F + I \cdot B \log B))$
- The computational cost of ITI is $O(F + I \cdot T)$ where $T$ is the number of logistic regression steps
- The computational cost of Lin-AcT is $O(2F + I \cdot B \log B)$

**Memory required during training:**

- $M$: Memory required by the model activations during the forward pass
- $L$: Memory required to compute the forward pass on one layer
- $D$: Affine parameter weight matrix size

**Memory Requirements per Method:**

- LinEAS: $M + ID$
- LinAcT: $L + D$
- ITI: $L + D$

## C  User Study on LMs

We conducted a user preference study to evaluate the perceived quality of continuations generated by different intervention methods. The study involved a pool of 20 volunteers. Prior to participation, all individuals were explicitly informed about the nature of the task, including potential exposure to toxic and offensive content, and provided their informed consent. Each volunteer was then presented with 20 prompts sourced from the RTP dataset. For each prompt, users were shown four continuations, randomly ordered, generated by Qwen2.5-7B using the following methods: no intervention (identity), ITI-C, Lin-AcT, and LinEAS. The annotators' task was to select the continuation that was both least toxic and most coherent overall.

The aggregated results of the preference study are summarized in the table below, showing the percentage of times each method was preferred, along with its standard deviation. Our findings indicate a strong preference for LinEAS over three other alternatives: Linear-ACT, ITI-C, and no intervention (identity). Specifically, users preferred LinEAS in 57.70% of cases.

Table 6: Human Preference Study Results

| Method | Identity (%) | ITI (%) | Lin-AcT (%) | LinEAS (%) |
|---|---|---|---|---|
| Preference | $12.19 \pm 5.5$ | $11.67 \pm 8.7$ | $18.43 \pm 11.49$ | $57.70 \pm 14.64$ |

# D  Optimization details for LinEAS

See Algo. 1 for the description of a single LinEAS optimization step. Recall that for a vector $z \in \mathbb{R}^d$, the proximal operators of the 1-norm (a.k.a. soft-thresholding) and 2-norm are given by:

$$\mathrm{ST}_\tau(z) := \mathrm{sign}(z) \odot \max(|z| - \tau, \mathbf{0}) \qquad \text{and} \qquad \mathrm{Prox}_{\tau\|\cdot\|_2}(z) := \left(1 - \frac{\tau}{\|z\|_2}\right)_+ z. \quad (7)$$

---

**Algorithm 1** Proximal E2E Training Step.

---

1: **Require:** prompts $(\boldsymbol{x}^i)_i \sim p$, $(\boldsymbol{y}^j)_j \sim q$, LR $\rho$.
2: (pre-) compute activations $\boldsymbol{\eta}_\ell^i, i \leq n, \ell \leq L$ $\qquad\qquad\qquad\qquad\qquad\qquad\qquad$ ▷ Eq.(2)
3: compute activations lists $\boldsymbol{\xi}_\ell^i, i \leq n, \ell \leq L$. $\qquad\qquad\qquad\qquad\qquad\qquad\qquad$ ▷ Eq.(1)
4: set loss to $\mathcal{C} = 0$
5: **for** $\ell \leq L$ **do** $\qquad\qquad\qquad\qquad\qquad\qquad\qquad\qquad\qquad\qquad\qquad\qquad\qquad$ ▷ Forward
6: $\qquad Z := [\boldsymbol{\xi}_\ell^1, \dots, \boldsymbol{\xi}_\ell^n] \in \mathbb{R}^{n \times d_\ell}$ $\qquad\qquad\qquad\qquad\qquad\qquad\qquad\qquad$ ▷ Eq. (1)
7: $\qquad V := [\boldsymbol{\eta}_\ell^1, \dots, \boldsymbol{\eta}_\ell^n] \in \mathbb{R}^{n \times d_\ell}$ $\qquad\qquad\qquad\qquad\qquad\qquad\qquad\qquad$ ▷ Eq. (2)
8: $\qquad \mathcal{C} \leftarrow \mathcal{C} + \Delta(Z, V)$ $\qquad\qquad\qquad\qquad\qquad\qquad\qquad$ ▷ $\ell$-layer loss, Eq. (4)
9: **end for**
10: **for** $\ell \leq L$ **do**
11: $\qquad g_\omega, g_b \leftarrow \nabla_{\omega_\ell, b_\ell} \mathcal{C}$ $\qquad\qquad\qquad\qquad\qquad\qquad\qquad\qquad$ ▷ Backpropagation
12: $\qquad \omega_\ell, b_\ell \leftarrow \omega_\ell - \rho\, g_\omega, b_\ell - \rho\, g_b$ $\qquad\qquad\qquad\qquad\qquad\qquad\qquad\qquad$ ▷ Updates
13: $\qquad \omega_\ell \leftarrow \mathrm{Prox}_{\gamma\lambda_G\|\cdot\|_2} \circ \mathrm{ST}_{\gamma\lambda_1}(\omega_\ell - \mathbf{1}) + \mathbf{1}$ $\qquad\qquad\qquad\qquad\qquad$ ▷ Eq. (7)
14: $\qquad b_\ell \leftarrow \mathrm{Prox}_{\gamma\lambda_G\|\cdot\|_2} \circ \mathrm{ST}_{\gamma\lambda_1}(b_\ell)$ $\qquad\qquad\qquad\qquad\qquad\qquad\qquad$ ▷ Eq. (7)
15: **end for**

---

# E  Toxicity Mitigation (extended results)

Table 7 is an extension of Table 1 in which we include one more model, DeepSeek-7B, and more baselines. Specifically we include here another oracle baseline, a LoRA adapter, that we named LRTC-RL, and that is trained similarly to LoFIT-RL as explained in the main paper. More details on the training of these strongly supervised baselines can be found in Appendix I.

Additionally we include here CAA as proposed in the original work (*i.e.,* intervening only on the middle layer rather than on all layers as shown in the main paper).

In Table 8 we show results analogous to those in Table 7 but in a higher data regime, *i.e.,* using 1024 sentences per set.

Additionally, in Table 9, we show how the different activation steering methods perform in the setting of Section 4.1, when intervening different layer types (namely `.*post_.*_layernorm` and `.*o_proj` of the models' Huggingface implementation). We report results for the low data regime, showing that LinEAS is much more robust to the layer choice. Indeed, for `.*post_.*_layernorm` and models DeepSeek-7B and Qwen2.5-1.5B, ITI-C and Lin-ACT induce a toxicity slightly higher than the original one (marked in red).

| Model | Method | #Params | $\text{Tox}_{\text{RTC}}^{\text{RTP}}$ ($\downarrow$) | $\text{Tox}_{\text{RTC}}^{\text{TET}}$ ($\downarrow$) | $\text{PPL}_{\text{WIK}}$ ($\downarrow$) | MMLU ($\uparrow$) |
|---|---|---|---|---|---|---|
| Q1.5B | None | - | $3.00_{0.54}$ | $23.09_{0.67}$ | $13.67_{0.00}$ | $60.95_{0.00}$ |
| | LRTC-RL | 0.54M | $1.07_{0.40}$ | $7.94_{0.01}$ | $13.70_{0.10}$ | $60.78_{0.17}$ |
| | LoFIT-RL | 0.8596M | $0.37_{0.06}$ | $4.36_{0.00}$ | $14.12_{0.07}$ | $59.74_{0.14}$ |
| | Prompt | - | $4.07_{0.38}$ | $21.02_{1.44}$ | $13.65_{0.00}$ | $60.96_{0.00}$ |
| | CAA (mid) | 0.0015M | $2.86_{0.53}$ | $23.33_{1.25}$ | $13.69_{0.02}$ | $60.47_{0.18}$ |
| | CAA | 0.043M | $1.15_{0.37}$ | $5.77_{2.14}$ | $\color{red}{19.30_{2.76}}$ | $\color{red}{37.67_{6.95}}$ |
| | ReFT | 0.39M | $2.57_{0.60}$ | $18.17_{3.04}$ | $15.58_{0.52}$ | $58.84_{0.23}$ |
| | ITI-C | 0.043M | $1.87_{0.21}$ | $18.16_{0.62}$ | $12.39_{0.09}$ | $60.88_{0.08}$ |
| | Lin-AcT | 0.086M | $1.50_{0.35}$ | $13.88_{1.72}$ | $13.88_{0.16}$ | $60.09_{0.25}$ |
| | LinEAS | 0.086M | $\mathbf{1.07}_{0.46}$ | $\mathbf{12.70}_{0.74}$ | $14.10_{0.07}$ | $59.97_{0.16}$ |
| G2-2B | None | - | $4.00_{0.45}$ | $13.39_{1.42}$ | $14.79_{0.00}$ | $53.03_{0.00}$ |
| | LRTC-RL | 0.8M | $0.83_{0.25}$ | $3.47_{0.01}$ | $15.38_{0.17}$ | $52.56_{0.11}$ |
| | LoFIT-RL | 0.1065M | $0.40_{0.20}$ | $1.76_{0.00}$ | $15.43_{0.08}$ | $52.17_{0.17}$ |
| | Prompt | - | $4.60_{0.36}$ | $12.32_{0.67}$ | $14.81_{0.00}$ | $53.18_{0.00}$ |
| | CAA (mid) | 0.0023M | $4.93_{0.42}$ | $14.04_{0.52}$ | $14.88_{0.02}$ | $51.49_{0.51}$ |
| | CAA | 0.06M | $0.80_{0.00}$ | $2.44_{1.99}$ | $\color{red}{23.52_{2.67}}$ | $\color{red}{26.86_{0.08}}$ |
| | ReFT | 0.54M | $2.85_{0.49}$ | $11.15_{1.91}$ | $\color{red}{19.93_{0.30}}$ | $48.99_{1.34}$ |
| | ITI-C | 0.06M | $1.17_{0.60}$ | $7.15_{0.92}$ | $14.00_{0.11}$ | $52.78_{0.23}$ |
| | Lin-AcT | 0.12M | $1.60_{0.32}$ | $7.76_{0.39}$ | $14.78_{0.12}$ | $52.43_{0.57}$ |
| | LinEAS | 0.24M | $\mathbf{0.73}_{0.10}$ | $\mathbf{4.02}_{0.68}$ | $15.46_{0.21}$ | $52.22_{0.40}$ |
| D7B | None | - | $4.30_{0.70}$ | $18.62_{0.51}$ | $8.49_{0.00}$ | $48.31_{0.00}$ |
| | LRTC-RL | 1.97M | $1.97_{0.38}$ | $5.07_{0.00}$ | $8.67_{0.05}$ | $47.76_{0.34}$ |
| | LoFIT-RL | 0.2458 | $0.53_{0.15}$ | $1.63_{0.00}$ | $9.31_{0.05}$ | $46.78_{0.14}$ |
| | Prompt | - | $4.20_{0.70}$ | $15.69_{0.82}$ | $8.51_{0.00}$ | $48.23_{0.00}$ |
| | CAA (mid) | 0.0043M | $4.72_{0.54}$ | $19.07_{0.98}$ | $8.73_{0.17}$ | $44.76_{1.98}$ |
| | CAA | 0.13M | $0.07_{0.15}$ | $0.33_{0.65}$ | $\color{red}{> 1000}$ | $\color{red}{23.14_{0.34}}$ |
| | ReFT | 1.11M | $2.25_{1.04}$ | $10.39_{6.01}$ | $\color{red}{51.58_{40.2}}$ | $\color{red}{35.56_{11.1}}$ |
| | ITI-C | 0.13M | $2.83_{0.40}$ | $15.18_{2.00}$ | $7.71_{0.07}$ | $48.47_{0.25}$ |
| | Lin-AcT | 0.25M | $\mathbf{2.23}_{0.69}$ | $\mathbf{11.08}_{0.76}$ | $8.67_{0.03}$ | $47.71_{0.27}$ |
| | LinEAS | 0.25M | $2.30_{0.14}$ | $12.09_{0.83}$ | $8.38_{0.05}$ | $48.13_{0.07}$ |
| Q7B | None | - | $3.92_{0.59}$ | $25.16_{0.92}$ | $10.67_{0.00}$ | $74.26_{0.00}$ |
| | LRTC-RL | 1.26M | $1.30_{0.44}$ | $6.59_{0.01}$ | $10.68_{0.06}$ | $74.08_{0.15}$ |
| | LoFIT-RL | 0.10M | $1.10_{0.38}$ | $7.11_{0.30}$ | $10.91_{0.16}$ | $73.87_{0.17}$ |
| | Prompt | - | $6.80_{0.00}$ | $21.22_{0.21}$ | $10.65_{0.00}$ | $74.23_{0.00}$ |
| | CAA (mid) | 0.0036M | $4.00_{0.45}$ | $22.32_{1.13}$ | $10.66_{0.03}$ | $73.45_{0.14}$ |
| | CAA | 0.10M | $1.20_{0.00}$ | $9.25_{3.07}$ | $12.83_{0.00}$ | $\color{red}{48.58_{0.00}}$ |
| | ReFT | 0.90M | $3.33_{0.96}$ | $20.38_{2.37}$ | $13.80_{1.20}$ | $70.43_{0.60}$ |
| | ITI-C | 0.10M | $2.63_{0.44}$ | $19.98_{1.24}$ | $9.63_{0.03}$ | $74.08_{0.05}$ |
| | Lin-AcT | 0.20M | $2.72_{0.46}$ | $21.64_{2.00}$ | $11.42_{0.34}$ | $72.18_{0.16}$ |
| | LinEAS | 0.20M | $\mathbf{1.95}_{0.48}$ | $\mathbf{14.95}_{0.92}$ | $10.91_{0.35}$ | $73.67_{0.05}$ |

Table 7: Toxicity mitigation on the RTP and TET datasets using four different models, Q1.5B: Qwen2.5-1.5B, G2-2B: Gemma2-2B, D7B: DeepSeek-7B and Q7B: Qwen2.5-7B. Strongly degraded utility is marked in red. We report results at low ($N = 32$ sentences to estimate the interventions) data regime. For each method, we use the best intervention layers according to an ablation study. See Table 8 in Appendix for an ablation with larger training size. Results for LinEAS improve significantly on ITI-C and Lin-AcT with similar impact on quality metrics. The quality of these interventions is often on par with the strong baselines LRTC-RL and LoFIT-RL, despite this approach being far more involved (both in terms of parameter size, access to ground truth labeling oracle RTC, and compute).

| Model | method | # par (M) | $\text{Tox}_{\text{RTC}}^{\text{RTP}}$ ($\downarrow$) | $\text{Tox}_{\text{RTC}}^{\text{TET}}$ ($\downarrow$) | $\text{PPL}_{\text{WIK}}$ ($\downarrow$) | MMLU ($\uparrow$) |
|---|---|---|---|---|---|---|
| Gemma2-2B | None | - | $4.00_{0.45}$ | $13.39_{1.42}$ | $14.79_{0.00}$ | $53.03_{0.00}$ |
| | LRTC-RL | 0.8M | $0.50_{0.44}$ | $1.68_{0.01}$ | $15.78_{0.19}$ | $52.45_{0.54}$ |
| | Prompt | - | $4.60_{0.36}$ | $12.32_{0.67}$ | $14.81_{0.00}$ | $53.18_{0.00}$ |
| | CAA (mid) | 0.0023M | $4.23_{0.72}$ | $12.41_{0.68}$ | $14.83_{0.00}$ | $52.32_{0.08}$ |
| | CAA | 0.06M | $0.70_{0.14}$ | $3.17_{0.74}$ | $16.38_{0.01}$ | $\textcolor{red}{46.44}_{0.02}$ |
| | ReFT | 0.54M | $3.73_{0.95}$ | $14.04_{1.61}$ | $15.40_{0.18}$ | $51.32_{0.25}$ |
| | ITI-c | 0.06 | $0.30_{0.26}$ | $2.68_{0.43}$ | $14.65_{0.06}$ | $52.04_{0.17}$ |
| | Lin-AcT | 0.12 | $1.07_{0.52}$ | $6.08_{0.67}$ | $14.85_{0.04}$ | $52.36_{0.11}$ |
| | LinEAS | 0.24 | $0.95_{0.26}$ | $3.46_{0.44}$ | $15.82_{0.02}$ | $51.28_{0.08}$ |
| DeepSeek-7B | None | - | $4.30_{0.70}$ | $18.62_{0.51}$ | $8.49_{0.00}$ | $48.31_{0.00}$ |
| | LRTC-RL | 1.97M | $0.90_{0.30}$ | $1.95_{0.00}$ | $8.80_{0.03}$ | $47.28_{0.58}$ |
| | Prompt | - | $4.20_{0.57}$ | $15.88_{0.90}$ | $8.51_{0.00}$ | $48.23_{0.00}$ |
| | CAA (mid) | 0.0043M | $4.63_{0.25}$ | $21.38_{1.29}$ | $8.72_{0.15}$ | $46.33_{0.48}$ |
| | CAA | 0.13M | $0.10_{0.14}$ | $0.08_{0.16}$ | $\textcolor{red}{>1000}$ | $\textcolor{red}{23.69}_{0.81}$ |
| | ReFT | 1.11M | $4.83_{0.78}$ | $17.76_{0.90}$ | $\textcolor{red}{12.34}_{0.65}$ | $\textcolor{red}{33.22}_{4.55}$ |
| | ITI-c | 0.13 | $1.77_{0.40}$ | $10.70_{1.14}$ | $7.79_{0.02}$ | $48.20_{0.15}$ |
| | Lin-AcT | 0.25 | $1.42_{0.43}$ | $9.39_{0.34}$ | $8.77_{0.01}$ | $47.74_{0.17}$ |
| | LinEAS | 0.25 | $1.70_{0.14}$ | $9.49_{0.52}$ | $8.53_{0.04}$ | $48.01_{0.04}$ |
| Qwen2.5-1.5B | None | - | $3.00_{0.54}$ | $23.09_{0.67}$ | $13.67_{0.00}$ | $60.95_{0.00}$ |
| | LRTC-RL | 0.54M | $0.67_{0.40}$ | $3.93_{0.01}$ | $13.91_{0.09}$ | $60.70_{0.19}$ |
| | Prompt | - | $4.07_{0.38}$ | $21.02_{1.44}$ | $13.65_{0.00}$ | $60.96_{0.00}$ |
| | CAA (mid) | 0.0015M | $2.87_{0.72}$ | $23.68_{1.04}$ | $13.67_{0.02}$ | $60.70_{0.10}$ |
| | CAA | 0.043M | $0.90_{0.24}$ | $6.69_{2.07}$ | $15.18_{1.02}$ | $\textcolor{red}{53.35}_{3.94}$ |
| | ReFT | 0.39M | $2.75_{0.29}$ | $14.33_{3.39}$ | $\textcolor{red}{35.48}_{20.8}$ | $\textcolor{red}{52.63}_{3.75}$ |
| | ITI-c | 0.043 | $1.60_{0.10}$ | $15.50_{0.81}$ | $12.53_{0.04}$ | $60.73_{0.21}$ |
| | Lin-AcT | 0.086 | $0.95_{0.38}$ | $11.61_{1.43}$ | $14.06_{0.03}$ | $59.82_{0.22}$ |
| | LinEAS | 0.086 | $0.90_{0.26}$ | $12.56_{0.70}$ | $14.20_{0.04}$ | $59.21_{0.16}$ |
| Qwen2.5-7B | None | - | $3.92_{0.59}$ | $25.16_{0.92}$ | $10.67_{0.00}$ | $74.26_{0.00}$ |
| | LRTC-RL | 1.26M | $1.50_{0.36}$ | $5.28_{0.00}$ | $11.03_{0.05}$ | $73.91_{0.10}$ |
| | Prompt | - | $6.40_{0.40}$ | $21.22_{0.21}$ | $10.65_{0.00}$ | $74.23_{0.00}$ |
| | CAA (mid) | 0.0036M | $3.88_{0.21}$ | $22.87_{0.54}$ | $10.64_{0.00}$ | $73.86_{0.04}$ |
| | CAA | 0.10M | $2.00_{0.00}$ | $11.24_{0.88}$ | $11.18_{0.00}$ | $68.59_{0.00}$ |
| | ReFT | 0.90M | $3.65_{1.32}$ | $22.70_{3.39}$ | $\textcolor{red}{17.42}_{3.21}$ | $\textcolor{red}{60.35}_{11.8}$ |
| | ITI-c | 0.10 | $2.33_{0.76}$ | $18.18_{2.00}$ | $9.66_{0.04}$ | $74.19_{0.10}$ |
| | Lin-AcT | 0.20 | $1.65_{0.26}$ | $13.60_{0.99}$ | $10.80_{0.02}$ | $73.60_{0.07}$ |
| | LinEAS | 0.20 | $1.52_{0.33}$ | $13.92_{0.54}$ | $10.89_{0.14}$ | $73.37_{0.07}$ |

Table 8: Toxicity mitigation on the RTP and TET datasets. We report results at high ($N = 1024$ sentences to estimate the interventions) data regime. We used 10k optimization steps, with mini-batches of size $n = 32$. In the high-data regime, Lin-AcT achieves an outstanding 0.68 $\text{Tox}_{\text{RTC}}^{\text{RTP}}$ for Gemma2-2B. However, this method struggles at reducing toxicity for the other models. Conversely, LinEAS achieves similar (Gemma2-2B) or better (other models) RTP toxicity mitigation than in the low data setup, and with better MMLU than Lin-AcT for all models. Similary, LinEAS outperforms all other methods on TET toxicity mitigation by a large margin.

| Model | Method | Layer | Data | $\lambda$ | $\text{Tox}_{\text{RTC}}^{\text{RTP}}$ ($\downarrow$) | $\text{Tox}_{\text{RTC}}^{\text{TET}}$ ($\downarrow$) | $\text{PPL}_{\text{WIK}}$ ($\downarrow$) | MMLU ($\uparrow$) |
|---|---|---|---|---|---|---|---|---|
| Gemma2-2B | None | .*post_.*_layernorm | - | - | $4.00_{0.45}$ | $13.39_{1.42}$ | $14.79_{0.00}$ | $53.03_{0.00}$ |
|  | ITI-C | .*post_.*_layernorm | 32 | 1.0 | $2.38_{0.91}$ | $10.00_{0.57}$ | $13.89_{0.13}$ | $52.92_{0.14}$ |
|  | Lin-AcT | .*post_.*_layernorm | 32 | 1.0 | $1.35_{0.17}$ | $7.32_{1.16}$ | $15.08_{0.13}$ | $51.52_{0.32}$ |
|  | LinEAS | .*post_.*_layernorm | 32 | 1.0 | $\mathbf{0.73}_{0.10}$ | $\mathbf{4.02}_{0.68}$ | $15.46_{0.21}$ | $52.22_{0.40}$ |
| DeepSeek-7B | None | .*post_.*_layernorm | - | - | $4.30_{0.70}$ | $18.62_{0.51}$ | $8.49_{0.00}$ | $48.31_{0.00}$ |
|  | ITI-C | .*post_.*_layernorm | 32 | 1.0 | $\textcolor{red}{7.23}_{0.76}$ | $\textcolor{red}{28.13}_{2.45}$ | $8.85_{0.40}$ | $45.82_{1.15}$ |
|  | Lin-AcT | .*post_.*_layernorm | 32 | 1.0 | $\textcolor{red}{5.62}_{0.25}$ | $\textcolor{red}{24.29}_{1.46}$ | $9.37_{0.20}$ | $45.95_{0.04}$ |
|  | LinEAS | .*post_.*_layernorm | 32 | 1.0 | $\mathbf{2.30}_{0.14}$ | $\mathbf{12.09}_{0.83}$ | $8.38_{0.05}$ | $48.13_{0.07}$ |
| Qwen2.5-1.5B | None | .*post_.*_layernorm | - | - | $3.00_{0.54}$ | $23.09_{0.67}$ | $13.67_{0.00}$ | $60.95_{0.00}$ |
|  | ITI-C | .*post_.*_layernorm | 32 | 1.0 | $2.62_{0.30}$ | $19.35_{1.52}$ | $13.23_{0.17}$ | $60.37_{0.24}$ |
|  | Lin-AcT | .*post_.*_layernorm | 32 | 1.0 | $2.75_{0.68}$ | $\textcolor{red}{25.51}_{1.79}$ | $16.33_{0.85}$ | $57.66_{0.56}$ |
|  | LinEAS | .*post_.*_layernorm | 32 | 1.0 | $\mathbf{1.07}_{0.46}$ | $\mathbf{12.70}_{0.74}$ | $14.10_{0.07}$ | $59.97_{0.16}$ |
| Qwen2.5-7B | None | .*post_.*_layernorm | - | - | $3.92_{0.59}$ | $25.16_{0.92}$ | $10.67_{0.00}$ | $74.26_{0.00}$ |
|  | ITI-C | .*post_.*_layernorm | 32 | 1.0 | $2.88_{0.60}$ | $19.41_{1.11}$ | $9.69_{0.07}$ | $74.13_{0.05}$ |
|  | Lin-AcT | .*post_.*_layernorm | 32 | 1.0 | $2.77_{0.39}$ | $20.57_{1.36}$ | $11.64_{0.24}$ | $72.21_{0.08}$ |
|  | LinEAS | .*post_.*_layernorm | 32 | 1.0 | $\mathbf{1.88}_{0.19}$ | $\mathbf{15.39}_{0.60}$ | $10.83_{0.25}$ | $73.56_{0.07}$ |
| Gemma2-2B | None | .*o_proj | - | - | $4.00_{0.45}$ | $13.39_{1.42}$ | $14.79_{0.00}$ | $53.03_{0.00}$ |
|  | ITI-C | .*o_proj | 32 | 0.5 | $\mathbf{1.17}_{0.60}$ | $\mathbf{7.15}_{0.92}$ | $14.00_{0.11}$ | $52.78_{0.23}$ |
|  | Lin-AcT | .*o_proj | 32 | 1.0 | $1.60_{0.32}$ | $7.76_{0.39}$ | $14.78_{0.12}$ | $52.43_{0.57}$ |
|  | LinEAS | .*o_proj | 32 | 1.0 | $2.10_{0.34}$ | $8.78_{0.56}$ | $14.72_{0.16}$ | $53.27_{0.41}$ |
| DeepSeek-7B | None | .*o_proj | - | - | $4.30_{0.70}$ | $18.62_{0.51}$ | $8.49_{0.00}$ | $48.31_{0.00}$ |
|  | ITI-C | .*o_proj | 32 | 0.5 | $2.83_{0.40}$ | $15.18_{2.00}$ | $7.71_{0.07}$ | $48.47_{0.25}$ |
|  | Lin-AcT | .*o_proj | 32 | 1.0 | $\mathbf{2.23}_{0.69}$ | $11.08_{0.76}$ | $8.67_{0.03}$ | $47.71_{0.27}$ |
|  | LinEAS | .*o_proj | 32 | 1.0 | $2.30_{0.48}$ | $\mathbf{8.46}_{0.54}$ | $8.61_{0.06}$ | $46.35_{0.37}$ |
| Qwen2.5-1.5B | None | .*o_proj | - | - | $3.00_{0.54}$ | $23.09_{0.67}$ | $13.67_{0.00}$ | $60.95_{0.00}$ |
|  | ITI-C | .*o_proj | 32 | 0.5 | $1.87_{0.21}$ | $18.16_{0.62}$ | $12.39_{0.09}$ | $60.88_{0.08}$ |
|  | Lin-AcT | .*o_proj | 32 | 1.0 | $\mathbf{1.50}_{0.35}$ | $13.88_{1.72}$ | $13.88_{0.16}$ | $60.09_{0.25}$ |
|  | LinEAS | .*o_proj | 32 | 1.0 | $\mathbf{1.50}_{0.29}$ | $\mathbf{12.03}_{0.71}$ | $14.04_{0.10}$ | $59.53_{0.17}$ |
| Qwen2.5-7B | None | .*o_proj | - | - | $3.92_{0.59}$ | $25.16_{0.92}$ | $10.67_{0.00}$ | $74.26_{0.00}$ |
|  | ITI-C | .*o_proj | 32 | 0.5 | $2.97_{0.21}$ | $20.68_{2.21}$ | $9.56_{0.02}$ | $74.20_{0.07}$ |
|  | Lin-AcT | .*o_proj | 32 | 1.0 | $2.25_{0.13}$ | $16.04_{0.95}$ | $10.77_{0.09}$ | $73.56_{0.08}$ |
|  | LinEAS | .*o_proj | 32 | 1.0 | $\mathbf{2.00}_{0.18}$ | $\mathbf{13.58}_{0.76}$ | $12.52_{0.07}$ | $71.34_{0.10}$ |

Table 9: **LinEAS is more robust to the layer choice.** Toxicity mitigation on the RTP and TET datasets, intervening on .*post_.*_layernorm and .*o_proj layers. We report results at low (32 sentences) regime, showing in **bold** the best toxicity result per model. When intervening on .*post_.*_layernorm, both ITI-C and Lin-AcT show poorer performance, specially for DeepSeek-7B and Qwen2.5-1.5B, where the toxicity goes even above the original one (in red). For .*o_proj, ITI-C and Lin-AcT perform better. Overall, LinEAS shows strong robustness to the layer choice.

# F  Order of Magnitude of Interventions Parameters

To inform the scale of regularization terms, we plot descriptive statistics of the values of **w** and **b**, layer by layer.

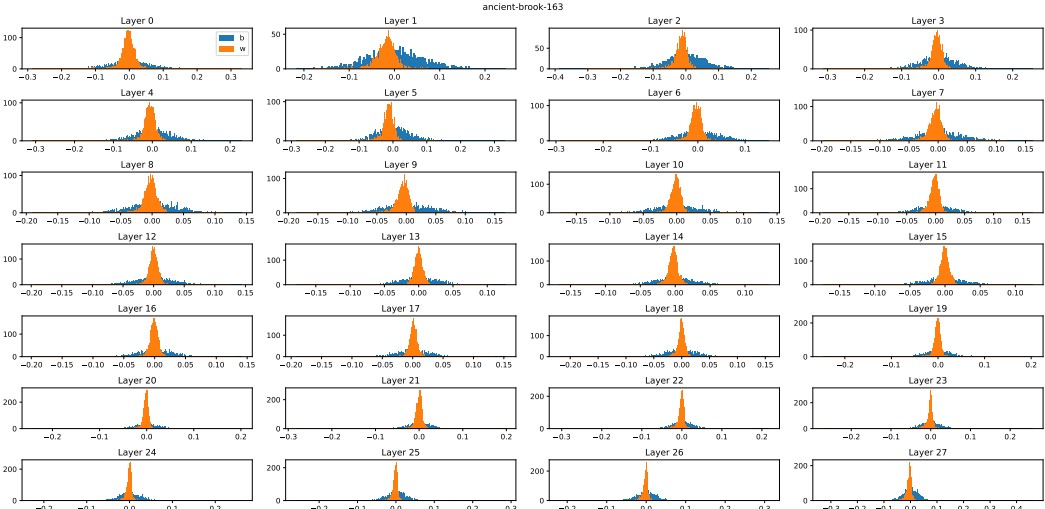

Figure 7: Distribution, layer by layer, of recentered scale parameters **w** and **b** biases, for a converged run of LinEAS, intervening on the 28 intervened layers of Gemma2-2B.

**Sparsity and Refitting.**  When $\gamma \gg 0$, several coordinates of $\omega_\ell$ and $b_\ell$ (parameters of $T_\ell$) will collapse to 1 or 0, respectively. While this is desired, non-zero parameters typically suffer from shrinking, where the regularization terms $\mathcal{R}_1, \mathcal{R}_G$ dampen the effect of $\mathcal{C}$. A typical solution to this phenomenon is to perform $m_{\text{post}}$ training steps updating only those non-collapsed parameters, a practice known as re-fitting in regression [38, 39]. We have not observed improvements when refitting parameters, and therefore do not use it. We observe that the entries of $\omega_\ell - \mathbf{1}$ and $b_\ell$ have similar scales (see Figure 7) and choose to use the same regularization strength.

# G  Effect of Sparsity on Toxicity Mitigation (extended results)

We complement the results shown in Figure 4, this time measuring the effect of sparsity on toxicity mitigation when the transport maps are optimized with 1024 (source and target) sentences. The results and trends discussed in Figure 4 also hold in this setup with more data.

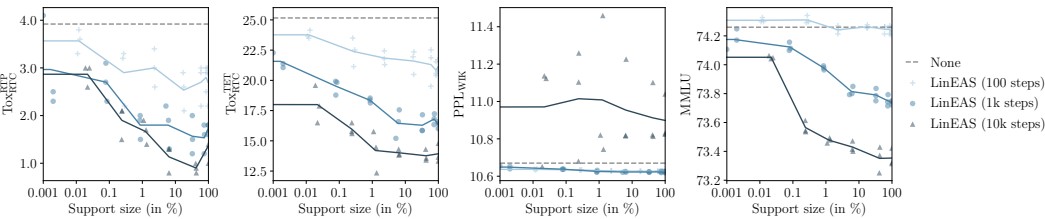

Figure 8: **Sparsity improves utility while mitigating toxicity, also in high data regime**. Toxicity results on Qwen2.5-7B using 1024 sentences, at different levels of sparsity $\gamma$ that result in different support sizes (x axis). With a support of 1%-5% we maintain similar toxicity (left, center-left) while $\text{PPL}_{\text{WIK}}$ decreases (center-right) and MMLU increases (right). Note that too long optimizations (30k steps) might harm utility, due to overfitting. Similarly, short optimizations (*e.g.,* 300 steps) and strong sparsity leads to low conditioning (mild toxicity mitigation).

## H   Effect of Sparsity on T2I Generation

We ran a sweep over sparsity coefficients on DMD2 [30] and report it on Table 10. We find that T2I models are *more* sensitive than LMs to the sparsity penalty, almost saturating to either full support or no support outside the range $[0.4, 0.8]$. We hypothesize the UNet is more sensitive than the transformer because its activation maps are less redundant due to the changes in dimensionality as described by Veit et al. [40] and Jastrzebski et al. [41].

Table 10: Effect of the sparsity coefficient on DMD2. Performance metrics (support, IMGScore, and CLIPScore) at varying levels of sparsity. IMGScore generally increases with higher sparsity, while CLIPScore shows a slight increase.

| sparsity | support(%) | IMGScore(%)↑ | CLIPScore(%)↓ |
|---|---|---|---|
| 0.0 | 100±0.0 | 71.4±5.5 | 13.1±3.2 |
| 0.4 | 92.7±5.0 | 84.7±6.0 | 14.5±2.9 |
| 0.5 | 76.7±14.0 | 89.5±6.3 | 15.0±2.9 |
| 0.6 | 51.6±22.1 | 94.8±3.6 | 15.5±2.8 |
| 0.7 | 13.8±11.4 | 98.9±1.1 | 16.0±2.8 |
| 0.8 | 3.3±3.2 | 99.6±0.4 | 16.1±2.8 |
| 1.0 | 0.0±0.0 | 100±0.0 | 16.1±2.9 |

## I   RLHF Implementation Details

We perform parameter-efficient adaptation of our baseline models with Huggingface's implementation of LoRA in their PEFT library and Huggingface's implementation of the PPO reinforcement learning algorithm in their TRL library. For each sample size in Table 1, we performed an hyperparameter search and chose the hyperparameters that yielded best validation toxicity scores at a perplexity close to LinEAS. Following Ouyang et al. [2], we fine-tune the models using proximal policy optimization (PPO) [42] on the same $N = 32$ data and use RTC (Roberta toxicity classifier) as our reward model. We instantiate the reward model with the Roberta toxicity classifier from Logacheva et al. [26] used for evaluation (RTC in Table 1); we use the base model without LoRA weights as the reference model and the LoRA model as the policy; we add an off-the-shelf value head from TRL to the policy to estimate the value function. We follow the original LoRA implementation and only fine-tune `{k,q,v,o}_proj` layers while keeping the MLPs frozen. The summary of hyperparameters can be found in Table 11.

| Hyperparameter | Values | Best 32 Samples | Best 1024 Samples |
|---|---|---|---|
| global_epochs | $\{10, 15, 20\}$ | 10 | 15 |
| ppo_epochs | $\{1, 10, 20, 50\}$ | 20 | 20 |
| learning_rate | $\{10^{-4}, 5 * 10^{-5}, 10^{-5}\}$ | $10^{-5}$ | $10^{-5}$ |
| batch_size | $\{32, 64, 128\}$ | 32 | 128 |
| mini_batch_size | $\{16, 32, 64\}$ | 32 | 64 |
| lora_rank | $\{2, 4, 8\}$ | 2 | 2 |

Table 11: List of hyper-parameters used to train PPO for 32 and 1024 samples. Unless otherwise specified, we use TRL's defaults. We could not use a mini-batch size greater than 64 due to memory constraints. We did not use gradient accumulation. We found the `learning_rate` and `global_epochs` to be the most important hyper-parameters. Low learning rate for few epochs leads to underfitting while high learning rate for many epochs tends to overfit.

## J   Composition of Interventions

We evaluate in this section the ability to compose two LinEAS maps pre-trained independently on two concepts. Our goal is to assess whether they can be composed, at the level of each activation, to induce *both* concepts. Achieving concept composition in activation steering is an open goal,and

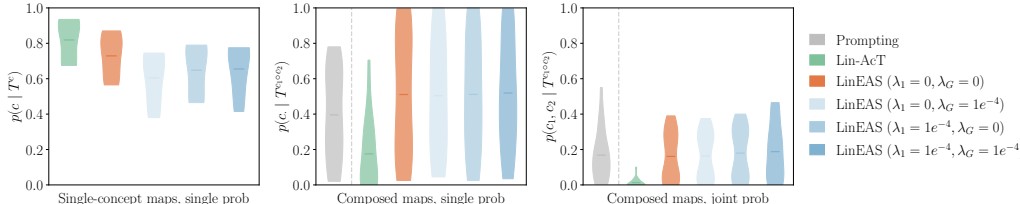

Figure 9: **LinEAS obtains composable maps.** We benchmark Lin-ACT and LinEAS for various regularization strengths $\gamma$ on a compositional task. We train each method to model distribution shifts towards a certain concept $c$ taking in a list of 5 concepts. **(Top-left)** We measure, using LLM as a judge, whether sentences formulated *using that trained shift* do contain that concept. We obtain for each concept a probability, that we then aggregate using violin plots. **(Top-middle)** We pick two concepts $c_1$ and $c_2$ randomly, train separately their maps, and compose them. We then measure the probability of finding $c_1$, and then $c_2$, in the generated content, using the composition of both maps (in both orders). **(Top-right)** We now measure the probability of finding simultaneously *both* concepts in the generations. As can be seen, the composition of LinEAS maps (learnt separately for 2 concepts) yields an average $15\times$ (right) higher probability of including both 2 concepts in the same generation, with respect to Lin-ACT, and a $3.1\times$ increase in generating at least one of the concepts (center). Interestingly, Lin-ACT obtains slightly better probability when using only single-concept maps (left), but these fail at composition. Additionally, note that sparsity in LinEAS is beneficial for compositionality, increasing the joint concept presence from 0.16 to 0.19 when using *group lasso*.

our hypothesis is that sparse and end-to-end LinEAS maps affect minimally the model, facilitating composisition.

**Setup.** For each of the concepts *day, night, elephant, football* and *fishing*, we generate $N = 50$ diverse sentences using Gemma2-27B that contain that concept, to form five target $(q_i)$. Additionally, we ask Gemma2-27B to generate 50 diverse sentences about generic situations, forming a single source distribution $p$. We then learn five steering maps, from $p$ to each of the $q_i$ distributions, using both Lin-ACT and LinEAS. For LinEAS we test all combinations of $\lambda_1 = 0, 1e^{-4}$ and $\lambda_G = 0, 1e^{-4}$, to assess the impact of sparsity. Equipped with these five concept maps, we compose them for each pair of concepts $c_1, c_2$ as follows: $T_\ell^{c_1 \circ c_2}(z) := T_\ell^{c_2}\big(T_\ell^{c_1}(z)\big)$. Note that $T_\ell^{c_1 \circ c_2} \neq T_\ell^{c_2 \circ c_1}$. Following Rodriguez et al. [10], we intervene on Gemma2-2B by generating 200 sentences that follow the prompt *Once upon a time* and we measure the presence of the concepts in the generations in a *LLM-as-a-judge manner*, querying Qwen2.5-7B-Instruct. More precisely, we query about: the presence of a concept $c$ in each generated sentence, when using the map trained with concept $c$; the presence of a concept $c$, when using the map trained with that concept *and* any other; the presence of both *two* concepts $c_1, c_2$ in the same sentence, yielding $p(c_1, c_2)$, using either $T_\ell^{c_1 \circ c_2}$ or $T_\ell^{c_2 \circ c_1}$.

**Results, comparing interventions.** Figure 9.(top) plots the three probabilities described above. In particular, the right plot shows $p(c_1, c_2 \mid T^{c_1 \circ c_2})$, *i.e.,* the probability of observing both concepts in the same generated sentence, using the composed map. Lin-ACT is able to generate concepts using single-concept maps (left plot) with average probability of 0.82 *vs.* 0.73 using LinEAS without sparsity. We also observe that increasing the sparsity (larger $\lambda$s) slightly diminishes the presence of concepts when using single-concept maps. However, we observe a drastically different picture when using *combined* maps: (middle) that probability goes from a Lin-ACT average of 0.17 to around 0.52 with LinEAS ($3.1\times$ increase). Most importantly, the joint presence of concepts probability (right) goes from 0.013 for Lin-ACT to 0.19 ($15\times$ increase) for LinEAS with *group lasso* regularization (both $\lambda_1, \lambda_G$ used). See Appendix J for generation examples. These results show that LinEAS learns maps that are easier to compose than those from Lin-ACT. Indeed, composition of Lin-ACT maps is very brittle. While LinEAS achieves much stronger compositionality, our results show that there is still room for improvement on this important problem. We provide qualitative examples in Figure 1 .

**Results, prompting as baseline.** We prompt the LLM to complete a generation with two concepts using *"Continue the following text, make sure concepts [c1] and [c2] appear in the continuation: Once upon a time"*. We observe that the mean probability of:

- generating either of the concepts (middle plot) is 39% with prompting / 52% for LinEAS.
- generating both concepts (right plot) is 17% for prompting / 19% for LinEAS.

In light of these results, we conclude that prompting is a strong baseline, since it has direct access to the concepts in textual form at generation time. However, LinEAS achieves better compositions without such direct access.

## J.1 Composition of Interventions: Qualitative Results

We show qualitative results using composed maps as detailed in Appendix J. In Table 12 and Table 13 we include generations that were marked as containing both concepts using LLM-as-a-judge (Qwen2.5-7B-Instruct). We compare Lin-AcT and LinEAS (with $\gamma = 1e^{-4}$).

Table 12: Generations inducing both concepts *Fishing*, *Elephant*. Only the provided 2 generations for Lin-AcT were marked as containing both concepts using LLM-as-a-judge, out of 200. Conversely, 25 sentences generated by LinEAS contain both concepts. Note the stark difference in quality, LinEAS achieves high quality generations with composition.

| Composition | Method | Generation |
|---|---|---|
| Fishing ∘ Elephant | Lin-AcT | Once upon a time, the Tuffa as the bull elephant's foot as she made a small putty as the water as she nosed the bank of the river as the water, as the sun as the swaying on the water as she stood in the shallows, the as the water [...] |
| Fishing ∘ Elephant | Lin-AcT | Once upon a time, a large, gray, as it was the bull as it snorted in the water as it struggled to break the line of the bank as it thrashed in the water, a small stream of water, as it made a final sprint, as the water, as it snorted noisily, the water as the river [...] |
| Fishing ∘ Elephant | LinEAS | Once upon a time, a man was fishing in the river when he saw a beautiful, pink baby elephant walking towards him. The elephant was thirsty and the man offered him a drink of water. The elephant was grateful and splashed some water on the man's face. |
| Fishing ∘ Elephant | LinEAS | Once upon a time, a huge elephant's tusk broke the water, his trunk splashing in the shallow river. "Hup!" he called, his tail swished against the muddy bank. I watched from the shore, my fishing rod dangling in the water. |
| Fishing ∘ Elephant | LinEAS | Once upon a time, the elephant's trunk broke the water, his massive body rising and disappearing. The jungle rumbled in the distance, their long tusks scraping against the mud. It was a young calf, its small, wet back. The fisherman sat on the bank, his net swinging lazily in the shallow water. A small fish |

Table 13: Generations inducing both concepts *Night*, *Football*.

| Composition | Method | Generation |
|---|---|---|
| Night ∘ Football | Lin-AcT | Once upon a time, a little girl's dream came true. It was the first night of the 2013-14 season and the young forward had just scored her first goal for the first team. The ball had nestled in the net in the 15th minute of the game, and the 16-year-old couldn't believe her luck. |
| Night ∘ Football | Lin-AcT | Once upon a time, there was a man who was so determined to win the game, he went out to play in the rain. It was a cold, wet night, and the rain was pouring down, but the man didn't let it dampen his spirits. He was out to win the game, and he knew that he had to make a late, late goal to seal the victory. |
| Night ∘ Football | Lin-AcT | Once upon a time, I was a happy camper. I was in the middle of a long drive through the woods, the sound of the wind whistling through the trees. The air was cool and the sun was setting, casting a warm glow over the forest. As I drove, I couldn't help but feel a sense of excitement. I had just won the race to the finish line, the ball bouncing off the net, and the ball was heading for the goal. |
| Night ∘ Football | LinEAS | Once upon a time, in the night, the lights of the field illuminated the players. The crowd roared in excitement, their voices echoing off the stands. It was the final whistle, and the opposing team celebrated, their coach shouting with joy. |
| Night ∘ Football | LinEAS | Once upon a time, the *"Let's go Rangers!"* could be heard through the dark, cold night. The home crowd cheered and the ball flew past the goal, sending a shower of confetti into the air. |
| Night ∘ Football | LinEAS | Once upon a time, the sun set on the field of grass. The opposing teams were locked in a fierce battle, the crowd roaring with excitement. It was the opening goal, and the referee blew his whistle, signaling the end of the game. |

## K   Group Sparsity Trade-offs

We consider the interventions learned with Gemma2-2B for the concepts *day*, *night*, *elephant*, *football*, and *fishing*, but set both $\lambda_1 > 0$ and $\lambda_G > 0$, enforcing a sparse group regularizer. Figure 10 shows how the proportion of intervened units, out of their entire support, is distributed across layers. Since we are plotting proportions of the support, differences between models in terms of the absolute number of impacted neurons are not reflected. Rather, the plot allows to grasp how regularization, and notably group regularization, impacts layer selection. An important finding of this experiment is that we do observe, for high group regularization regimes, that very specific layers are consistently selected to induce diverse concepts. Additionally, we posit that this grouping of neurons at certain layers impacts positively efficient transfer of concepts, notably when investigating compositionality.

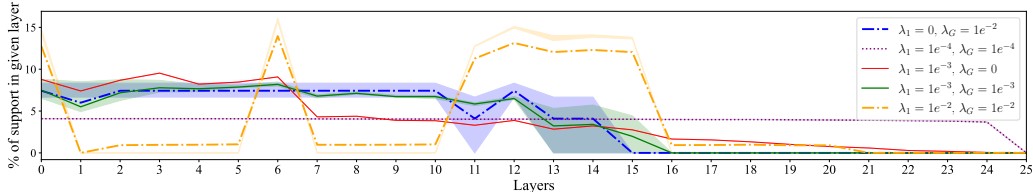

(a) LinEAS Distribution of support across layers (biases). Post Feed-Forward Layer Norms.

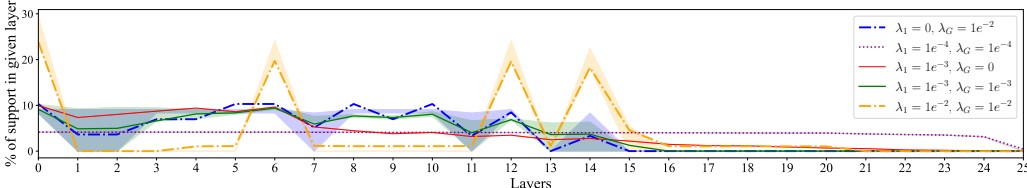

(b) LinEAS Distribution of support across layers (biases). Post Attention Layer Norms.

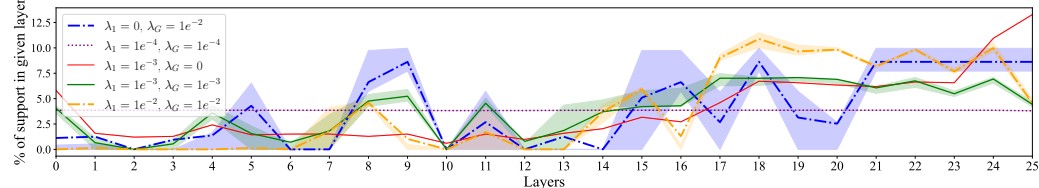

(c) LinEAS Distribution of support across layers (weights). Post Feed-Forward Layer Norms.

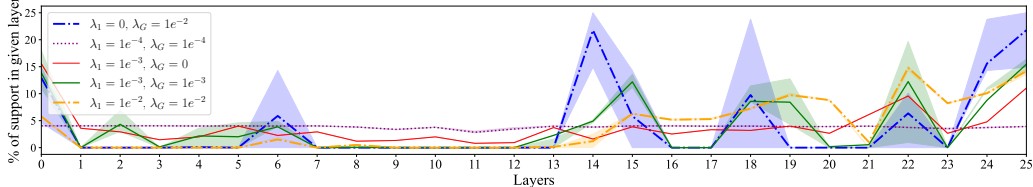

(d) LinEAS Distribution of support across layers (weights). Post Attention Layer Norms.

Figure 10: **LinEAS Distribution of support across layers.** Each subfigure shows the percentage of intervened units (out of the total support) across layers for different regularization strengths, averaged over 5 concepts (with 50% quantile range).

## L   Similarity of LinEAS Interventions with Human Judgment

We draw a set of 50 concepts, from the MEN dataset [29], a resource of 3,000 word pairs annotated with human similarity judgments (see also Fedzechkina et al. [43] for a study on LLM interpretability building on the same resource). With these 50 concepts, we can recover 20 word pairs annotated for their similarity in the MEN dataset. We train LinEAS interventions for each concept on Gemma2-2B. We then ask: do we recover similar interventions for similar concepts, and does sparsity help? We can answer positively to both: highly similar concepts have highly similar interventions; and enforcing sparsity through our scheme improves that correlation.

To measure this, we compute the average sparse support of interventions (shown as y-axis in Figure 11). We compute the similarity between the intervention vectors for each word pair, focusing on biases first: $s_{int}^b = \{\text{sim}(\boldsymbol{b}_{c1}, \boldsymbol{b}_{c2})\}_{\forall c1 \neq c2}$ (on the left in Figure 11). Finally, we consider the human similarity judgments $s_{hum}$ reported in the MEN dataset for each word pair, and compute their correlation with the intervention similarity, $\text{corr}(s_{int}^b, s_{hum})$. We do the same for the weights, in this case through $s_{int}^w$ (on the right in Figure 11 and ). The numbers in the scatterplots in Figure 11 correspond to the LinEAS regularization parameters $\lambda_1, \lambda_G$.

Note that sparsity helps improve correlation beyond the non-sparse version of LinEAS (noted as 0;0, with a support of 100% in Figure 11). Overall, LinEAS shows a strong correlation with human judgment.

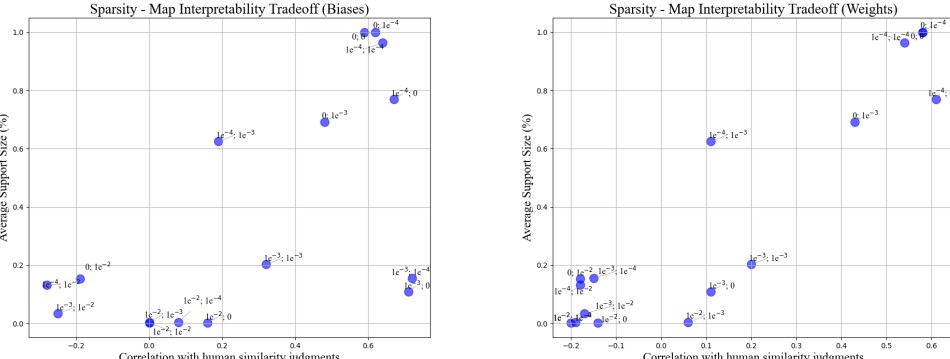

Figure 11: Scatter plots showing the correlation of interventions with human similarity judgment. We show the correlation with bias similarity in the left plot and with weight similarity on the right plot.

In Figure 12 we show the same results in form of matrix, showing that sparsity is indeed helpful to improve correlation with human alignment.

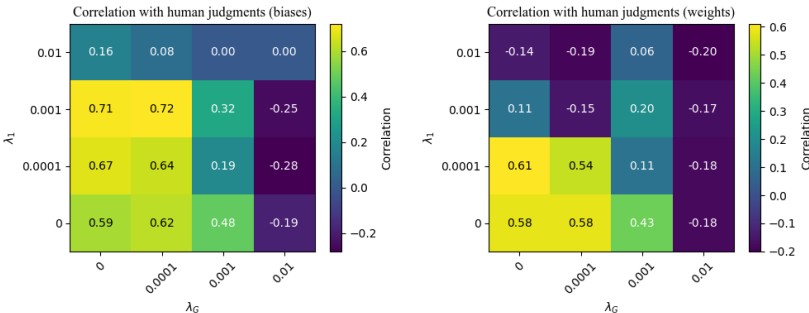

Figure 12: Heatmaps aggregating scatterplot results above but with a different view. We observe that highest correlations are obtained with a suitable regularization strength, for biases $\lambda_1 = 10^{-3}, \lambda_G = 10^{-4}$, for weights $\lambda_1 = 10^{-4}, \lambda_G = 0$.

## M   Additional Experiment: Inducing Truthfulness in LLMs

In this section, we complement and corroborate our insights from the experiments on toxicity mitigation in Section 4.1 with additional experiments on inducing truthfulness in LLMs, using the TruthfulQA benchmark [44]. In particular, we investigate how well LinEAS achieves to induce truthfulness on this benchmark in comparison to Lin-AcT, its strongest activation steering competitor from Section 4.1.

For LinEAS, we apply the intervention again to the post layernorm layers, while for Lin-AcT, we apply them to all layernorm layers as this was reported as optimal for Lin-AcT for TruthfulQA experiments in Rodriguez et al. [10]. We use 2-fold cross-validation on the 817 questions of the multiple choice part of the benchmark and learn the intervention on the concatenation of training fold questions concatenated with either incorrect (source) or correct (target) multiple-choice answer options. We report both MC1 and MC2 of TruthfulQA, and monitor overfitting on the TruthfulQA task by also evaluating MMLU 5-shot accuracy [28].

The results can be found in Table 14. We see that both methods can successfully induce truthfulness when presented with enough samples to learn the interventions, increasing the accuracy by up to

almost $5\%$ ($7\%$) on MC1 (MC2) in the high sample regime when 1024 samples are available. Overall the highest increases can be achieved with Lin-AcT, but only for the high sample regime. In the low sample regime, where only 32 samples are available, Lin-AcT tends to fail catastrophically: either it gets lower accuracies on MC1 and MC2 than even the unintervened model (Qwen2.5-7B, Qwen2.5-1.5B), or it fails completely on MMLU (Gemma2-2B). LinEAS on the other hand does well also in this low sample regime, and achieves second best overall performance on the Qwen models with only 32 samples available to learn interventions.

| Model | Samples | Method | MC1 Acc. (%) ($\uparrow$) | MC2 Acc. (%)($\uparrow$) | MMLU ($\uparrow$) |
|---|---|---|---|---|---|
| Qwen2.5-7B | - | None | $37.82_{0.00}$ | $52.14_{0.00}$ | $74.26_{0.00}$ |
| | 32 | Lin-AcT | $32.17_{0.66}$ | $47.69_{0.87}$ | $59.22_{1.74}$ |
| | | LinEAS | $\underline{40.10}_{0.37}$ | $\underline{55.74}_{0.31}$ | $\mathbf{73.88}_{0.07}$ |
| | 1024 | Lin-AcT | $\mathbf{42.59}_{0.50}$ | $\mathbf{58.92}_{0.82}$ | $73.79_{0.14}$ |
| | | LinEAS | $39.56_{0.13}$ | $55.20_{0.15}$ | $\underline{73.88}_{0.02}$ |
| Qwen2.5-1.5B | - | None | $30.23_{0.00}$ | $43.70_{0.00}$ | $60.95_{0.00}$ |
| | 32 | Lin-AcT | $27.22_{1.38}$ | $42.64_{3.17}$ | $32.10_{1.95}$ |
| | | LinEAS | $\underline{32.26}_{0.65}$ | $\underline{46.07}_{0.63}$ | $\underline{60.34}_{0.12}$ |
| | 1024 | Lin-AcT | $\mathbf{32.90}_{0.36}$ | $\mathbf{47.17}_{0.61}$ | $60.17_{0.18}$ |
| | | LinEAS | $31.77_{0.11}$ | $45.31_{0.20}$ | $\mathbf{60.41}_{0.01}$ |
| Gemma2-2B | - | None | $21.18_{0.00}$ | $33.05_{0.00}$ | $53.03_{0.00}$ |
| | 32 | Lin-AcT | $\underline{24.94}_{1.15}$ | $\mathbf{40.95}_{1.29}$ | $27.59_{1.09}$ |
| | | LinEAS | $24.21_{0.85}$ | $38.09_{1.14}$ | $\underline{52.08}_{0.16}$ |
| | 1024 | Lin-AcT | $\mathbf{25.65}_{0.53}$ | $\underline{39.73}_{0.49}$ | $51.40_{0.17}$ |
| | | LinEAS | $23.82_{0.19}$ | $37.80_{0.53}$ | $\mathbf{52.37}_{0.05}$ |

Table 14: Results on TruthfulQA. We report results at low data (32 samples to estimate the interventions) and high (1024 samples) regimes. Results are averaged over five random seeds.

# N   Additional details and results on text to image generation

## N.1   Additional qualitative results with LLM-generated prompts

We extend here the qualitative results mitigating and inducing styles on the 15 concepts described in Appendix O. Figures 13 to 15 compare Lin-AcT, and LinEAS. Both the diversity and (human) perceptual quality of the generations is higher with LinEAS. We observe stark differences between both methods' generations. Figures 16 to 18 show additional generations with LinEAS with more granular strengths.

Note that all the generations start from the same prompt, with the concept of interest appended in the form of textual tags. It is interesting to see that LinEAS recovers an image conforms with the prompt *without the concept* at $\lambda = 1$. Also, observe how LinEAS's generations are much more gradual than Lin-AcT.

We also comment on the surprising results obtained when inverting the LinEAS linear maps. We observe how the concept increases, and LinEAS shows much higher quality and coherence under this regime. This points to the fact that LinEAS is better exploiting the underlying structure in activation space.

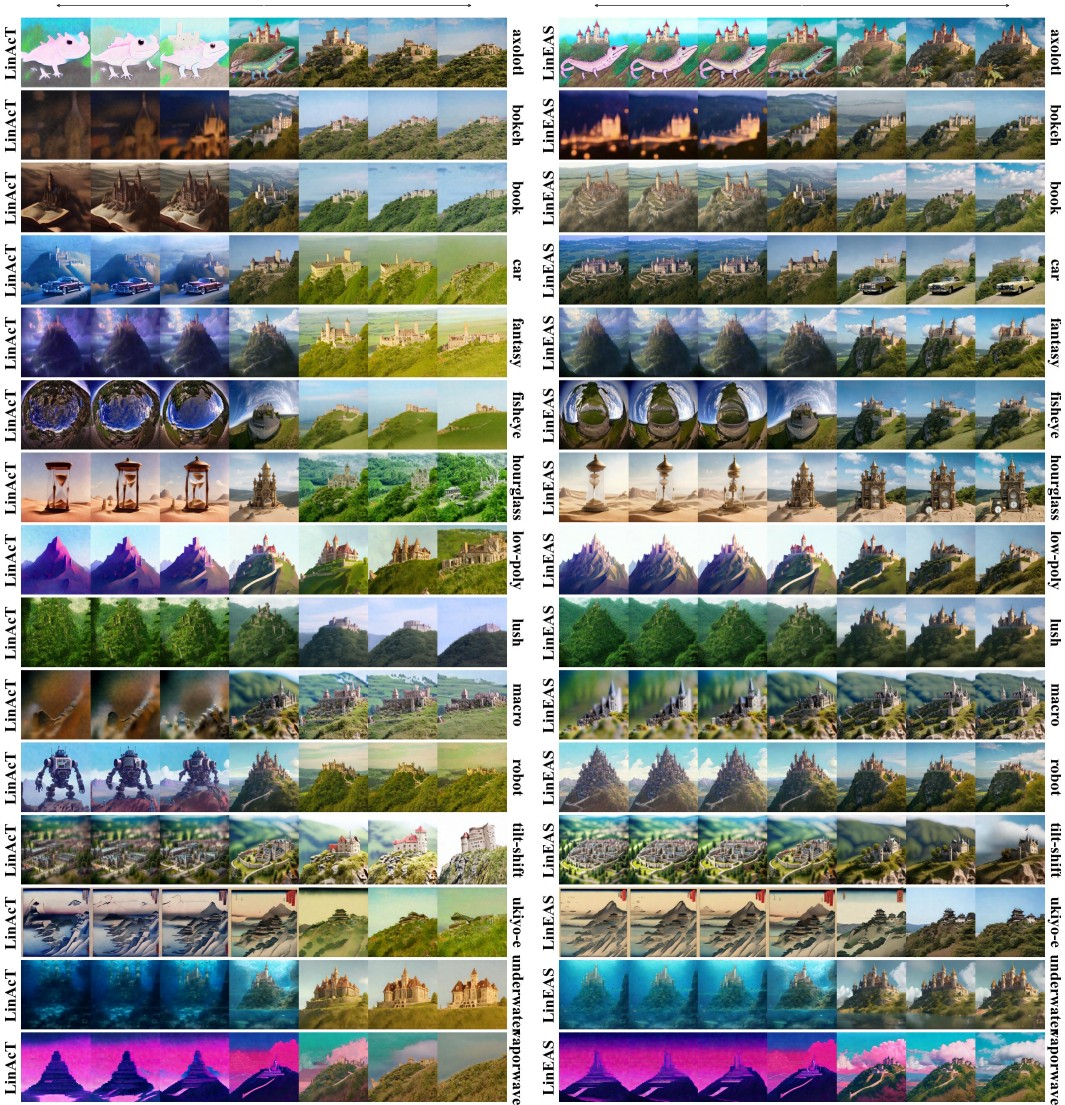

Figure 13: Images generated with Lin-AcT (left panel) and LinEAS (right panel) with the prompt *A grand castle sits atop a hill overlooking a valley. [concept tag]* Each row contains a different conditioning concept to be mitigated by the steering method and each column a different intervention strength ($\lambda$). Each column contains a generation for $\lambda = 1.0, 0.8, 0.6, 0, 0.6, 0.8, 1.0$ respectively and the columns to the left of "original" contain generations using $T_\ell^{-1}(z)$.

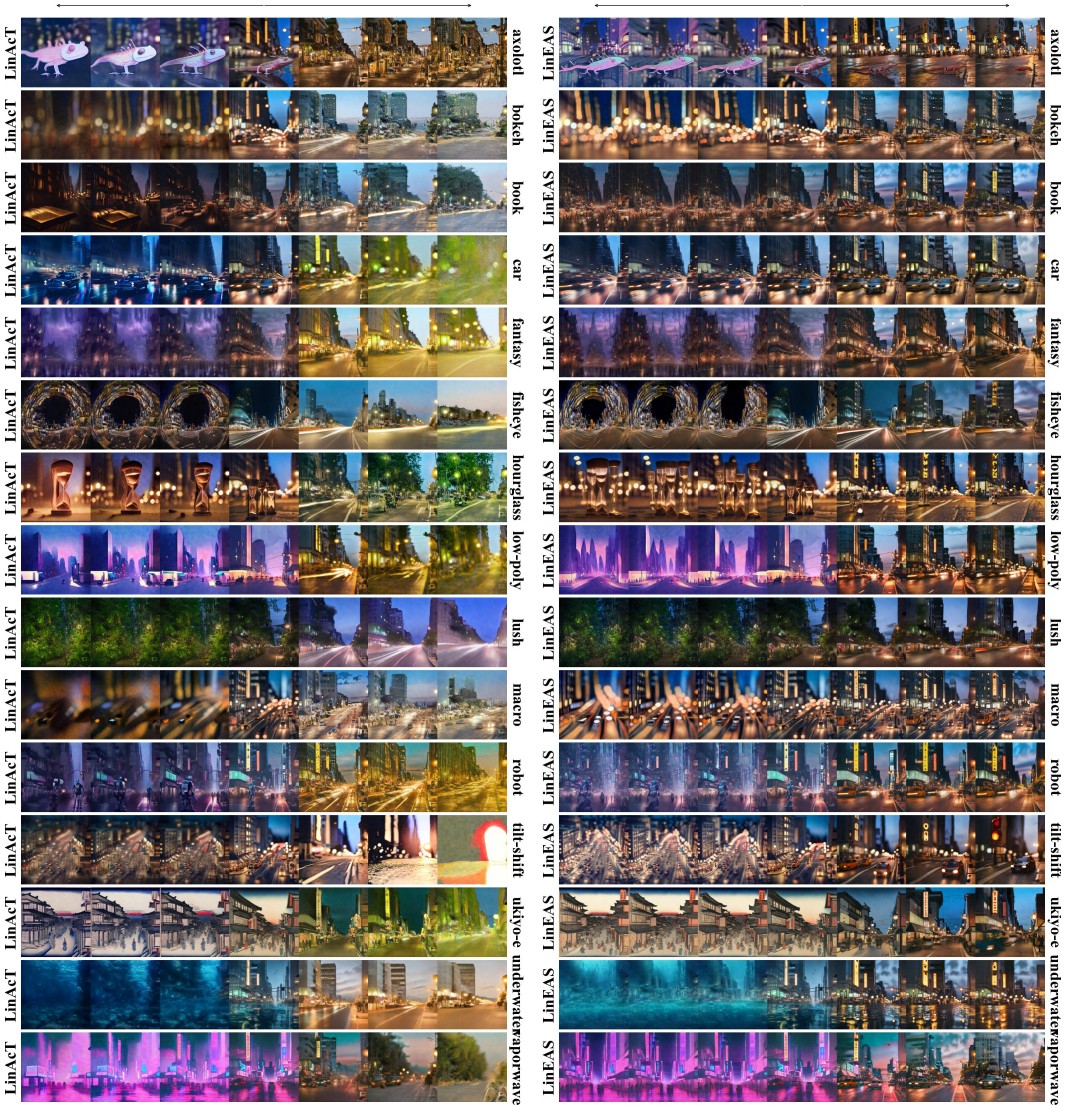

Figure 14: Images generated with Lin-AcT (left panel) and LinEAS (right panel) with the prompt *A bustling city street at twilight, lights blurring. [concept tag]* Each row contains a different conditioning concept to be mitigated by the steering method and each column a different intervention strength ($\lambda$). Each column contains a generation for $\lambda = [1.0, ..., 0, ..., 1.0]$ respectively and the columns to the left of "original" contain generations using $T_\ell^{-1}(z)$.

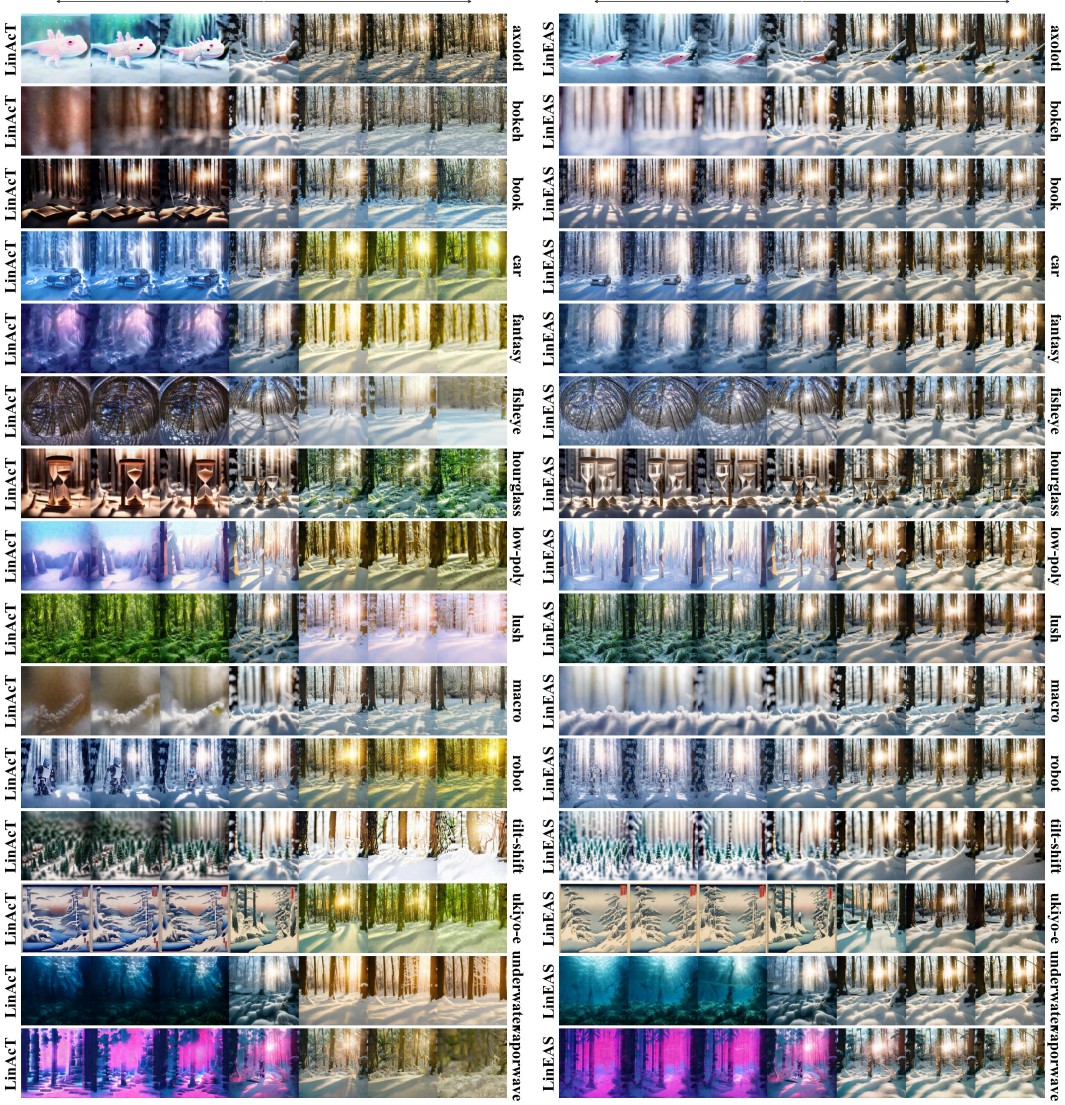

Figure 15: Images generated with Lin-AcT (left panel) and LinEAS (right panel) with the prompt *A snow-covered forest with sunlight filtering through the trees. [concept tag]* Each row contains a different conditioning concept to be mitigated by the steering method and each column a different intervention strength ($\lambda$). Each column contains a generation for $\lambda = [1.0, ..., 0, ..., 1.0]$ respectively and the columns to the left of "original" contain generations using $T_\ell^{-1}(z)$.

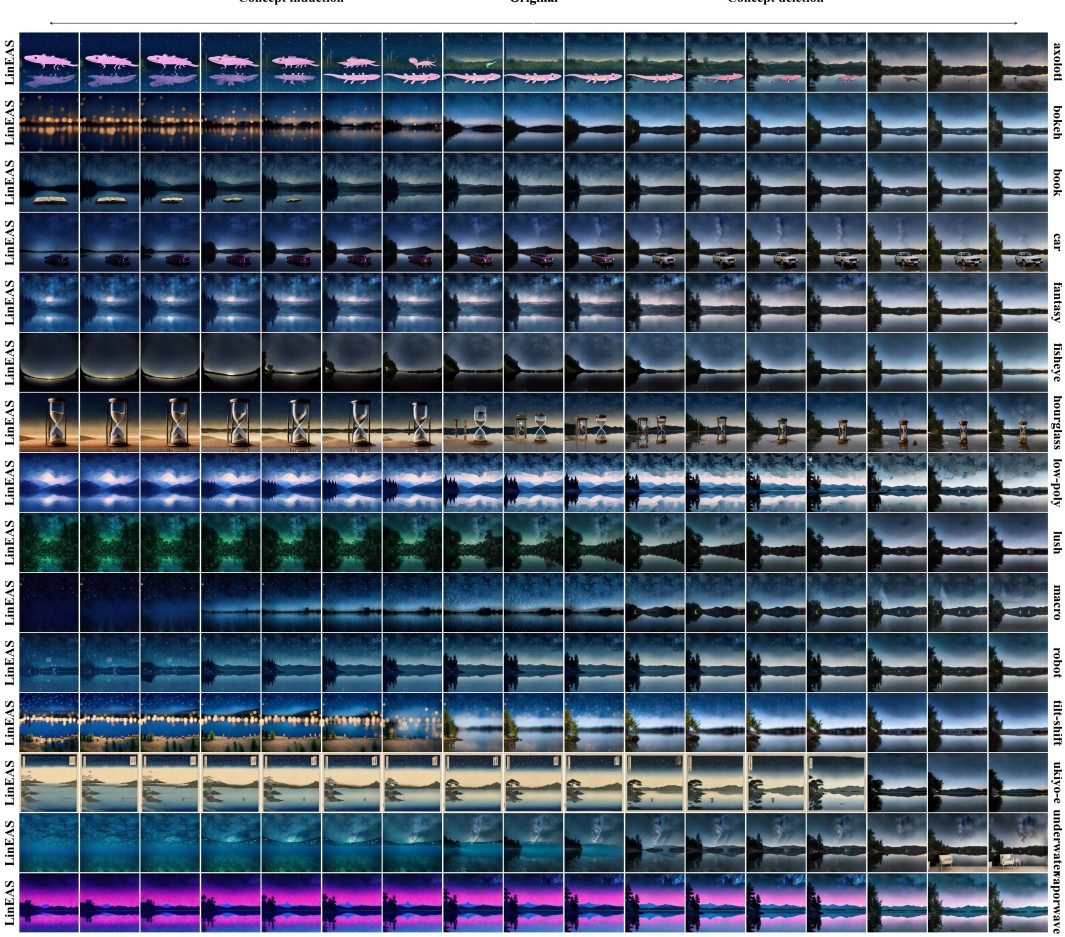

Figure 16: Images generated with LinEAS with the prompt *A starry night sky over a calm lake. [concept tag]* Each column contains images generated with a different steering strength for mitigating the concept corresponding to the row: $\lambda = [1.0, ..., 0, ..., 1.0]$. Images to the left of "original" were produced using the inverse steering map $T_\ell^{-1}(z)$.

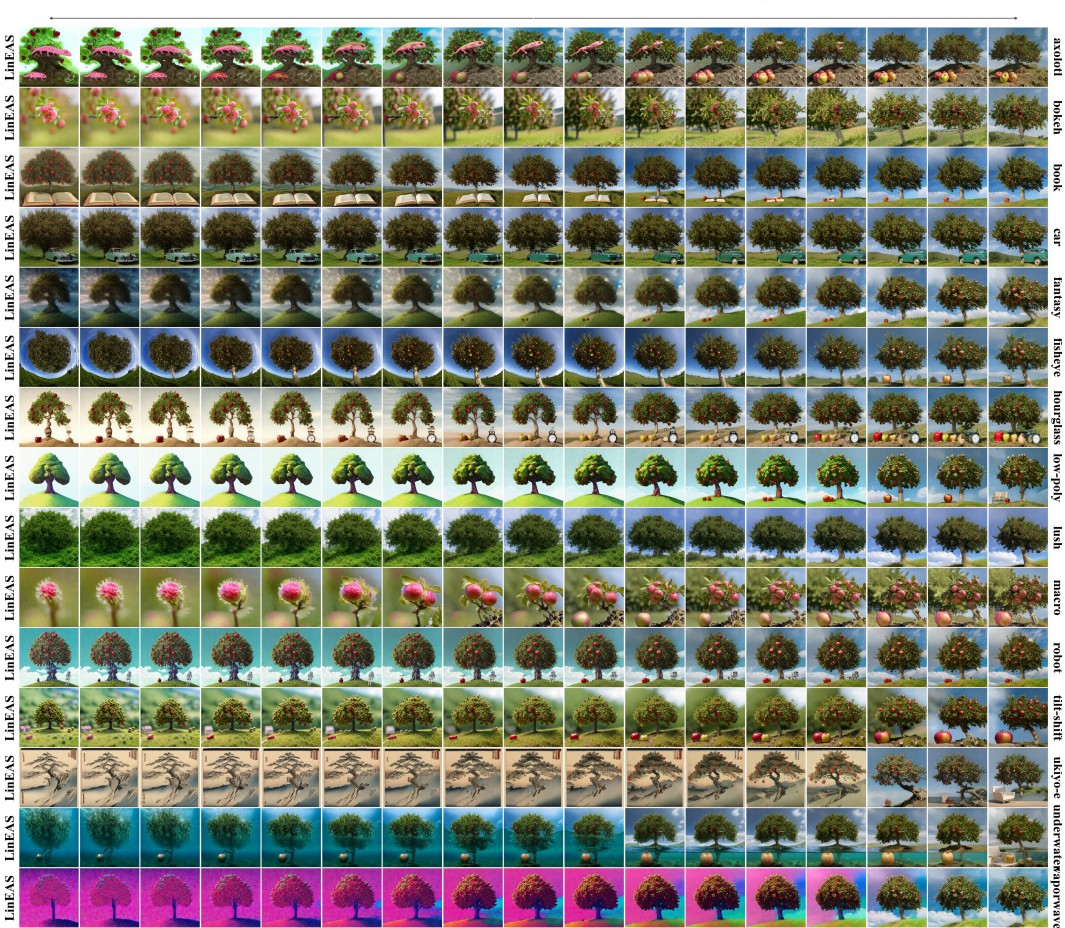

Figure 17: Images generated with LinEAS with the prompt *A fruiting apple tree on top of a hill. [concept tag]* Each column contains images generated with a different steering strength for mitigating the concept corresponding to the row: $\lambda = [1.0, ..., 0, ..., 1.0]$. Images to the left of "original" were produced using the inverse steering map $T_{\ell}^{-1}(z)$.

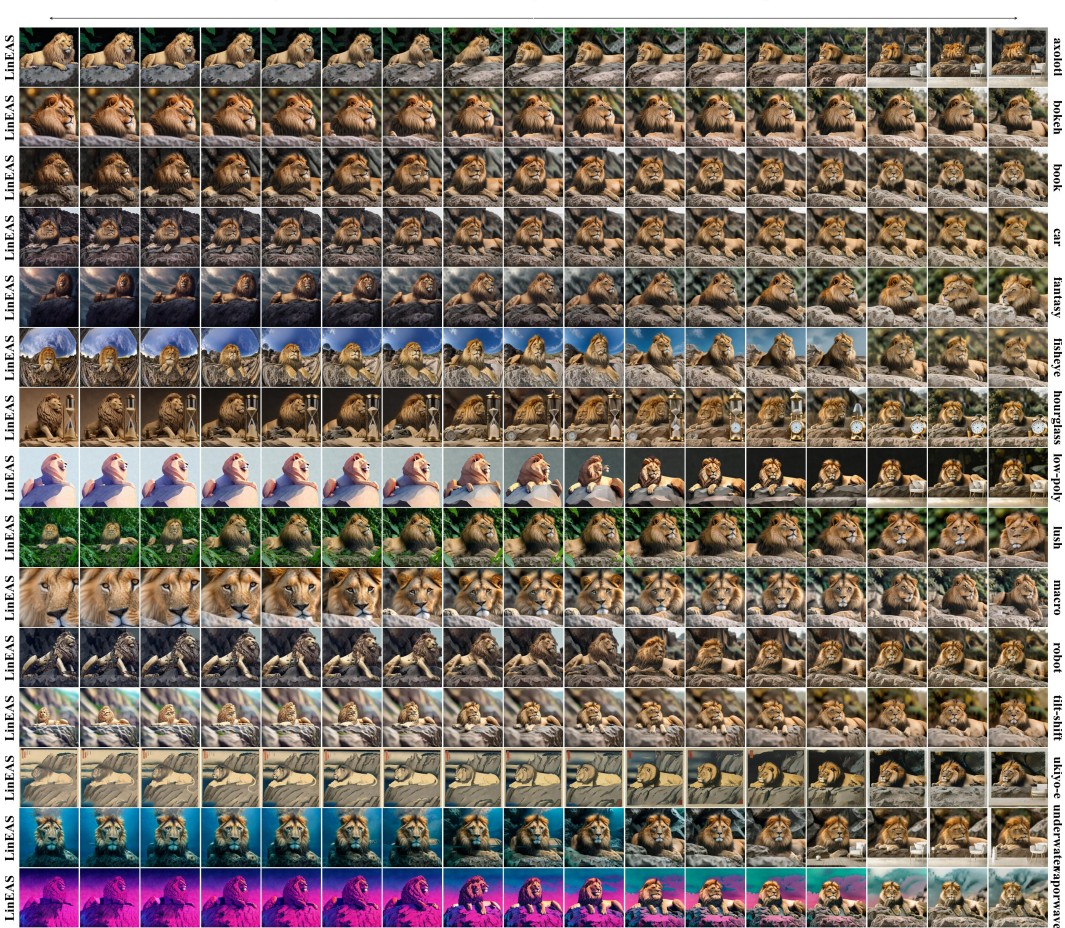

Figure 18: Images generated with LinEAS with the prompt *A majestic lion rests on a rocky outcrop.*
*[concept tag]* Each column contains images generated with a different steering strength for mitigating
the concept corresponding to the row: $\lambda = [1.0, ..., 0, ..., 1.0]$. Images to the left of "original" were
produced using the inverse steering map $T_\ell^{-1}(z)$.

## N.2 Detailed Quantitative Results

Here we report detailed per-concept ImgScores and ClipScores for ITI-C, Lin-AcT, and LinEAS both for the forward (($T_\ell(z)$)) and inverse (($T_\ell^{-1}(z)$)) application of the steering (Table 15). Remarkably, we find that inverting the steering operation tends to induce the concepts that these methods mitigate when they are not inverted.

| Task | Intervention | $T_\ell(z)$ ↑ImgScore | ↑ClipScore | $T_\ell^{-1}(z)$ ↑ImgScore | ↓ClipScore. |
|---|---|---|---|---|---|
| Axolotl | ITI-C | $0.13_{0.11}$ | $\mathbf{0.17}_{0.02}$ | $0.26_{0.17}$ | $0.27_{0.02}$ |
| | Lin-AcT | $0.35_{0.19}$ | $0.15_{0.02}$ | $0.28_{0.18}$ | $0.29_{0.03}$ |
| | LinEAS | $\mathbf{0.56}_{0.23}$ | $0.16_{0.04}$ | $\mathbf{0.60}_{0.19}$ | $\mathbf{0.23}_{0.05}$ |
| Bokeh | ITI-C | $0.38_{0.19}$ | $\mathbf{0.18}_{0.02}$ | $0.33_{0.18}$ | $\mathbf{0.24}_{0.02}$ |
| | Lin-AcT | $0.55_{0.18}$ | $\mathbf{0.18}_{0.02}$ | $0.31_{0.19}$ | $\mathbf{0.24}_{0.02}$ |
| | LinEAS | $\mathbf{0.75}_{0.14}$ | $\mathbf{0.18}_{0.02}$ | $\mathbf{0.62}_{0.22}$ | $\mathbf{0.24}_{0.02}$ |
| Book | ITI-C | $0.37_{0.20}$ | $\mathbf{0.19}_{0.01}$ | $0.45_{0.19}$ | $0.21_{0.02}$ |
| | Lin-AcT | $0.62_{0.16}$ | $0.18_{0.01}$ | $0.49_{0.19}$ | $0.22_{0.01}$ |
| | LinEAS | $\mathbf{0.80}_{0.10}$ | $0.17_{0.01}$ | $\mathbf{0.78}_{0.14}$ | $\mathbf{0.19}_{0.02}$ |
| Car | ITI-C | $0.18_{0.17}$ | $\mathbf{0.19}_{0.02}$ | $0.43_{0.23}$ | $0.24_{0.02}$ |
| | Lin-AcT | $0.37_{0.21}$ | $0.18_{0.02}$ | $0.42_{0.23}$ | $0.23_{0.02}$ |
| | LinEAS | $\mathbf{0.78}_{0.13}$ | $0.18_{0.03}$ | $\mathbf{0.81}_{0.12}$ | $\mathbf{0.19}_{0.03}$ |
| Fantasy | ITI-C | $0.12_{0.14}$ | $\mathbf{0.20}_{0.01}$ | $0.35_{0.19}$ | $\mathbf{0.21}_{0.02}$ |
| | Lin-AcT | $0.49_{0.19}$ | $\mathbf{0.20}_{0.02}$ | $0.52_{0.17}$ | $0.23_{0.01}$ |
| | LinEAS | $\mathbf{0.72}_{0.16}$ | $0.18_{0.02}$ | $\mathbf{0.73}_{0.15}$ | $\mathbf{0.21}_{0.02}$ |
| Fisheye | ITI-C | $0.40_{0.16}$ | $\mathbf{0.20}_{0.02}$ | $0.21_{0.13}$ | $0.25_{0.01}$ |
| | Lin-AcT | $0.49_{0.18}$ | $\mathbf{0.20}_{0.02}$ | $0.26_{0.15}$ | $0.25_{0.02}$ |
| | LinEAS | $\mathbf{0.68}_{0.14}$ | $\mathbf{0.20}_{0.02}$ | $\mathbf{0.51}_{0.16}$ | $\mathbf{0.24}_{0.02}$ |
| Hourglass | ITI-C | $0.16_{0.18}$ | $\mathbf{0.20}_{0.02}$ | $0.57_{0.26}$ | $0.30_{0.01}$ |
| | Lin-AcT | $0.29_{0.23}$ | $0.17_{0.02}$ | $0.60_{0.27}$ | $0.29_{0.01}$ |
| | LinEAS | $\mathbf{0.54}_{0.23}$ | $\mathbf{0.20}_{0.04}$ | $\mathbf{0.74}_{0.19}$ | $\mathbf{0.27}_{0.03}$ |
| Low-poly | ITI-C | $0.12_{0.12}$ | $\mathbf{0.19}_{0.02}$ | $0.50_{0.17}$ | $\mathbf{0.23}_{0.02}$ |
| | Lin-AcT | $0.39_{0.16}$ | $\mathbf{0.19}_{0.02}$ | $0.57_{0.14}$ | $0.24_{0.02}$ |
| | LinEAS | $\mathbf{0.57}_{0.19}$ | $0.18_{0.01}$ | $\mathbf{0.67}_{0.14}$ | $0.25_{0.02}$ |
| Lush | ITI-C | $0.29_{0.21}$ | $0.20_{0.02}$ | $0.44_{0.17}$ | $0.26_{0.01}$ |
| | Lin-AcT | $0.50_{0.20}$ | $\mathbf{0.21}_{0.02}$ | $0.47_{0.17}$ | $0.25_{0.01}$ |
| | LinEAS | $\mathbf{0.71}_{0.15}$ | $0.18_{0.02}$ | $\mathbf{0.75}_{0.12}$ | $\mathbf{0.23}_{0.02}$ |
| Macro | ITI-C | $0.31_{0.16}$ | $\mathbf{0.20}_{0.01}$ | $0.18_{0.15}$ | $\mathbf{0.22}_{0.02}$ |
| | Lin-AcT | $0.52_{0.18}$ | $0.18_{0.02}$ | $0.16_{0.13}$ | $\mathbf{0.22}_{0.02}$ |
| | LinEAS | $\mathbf{0.68}_{0.17}$ | $0.18_{0.02}$ | $\mathbf{0.54}_{0.21}$ | $0.23_{0.02}$ |
| Robot | ITI-C | $0.36_{0.20}$ | $\mathbf{0.20}_{0.02}$ | $0.32_{0.17}$ | $0.23_{0.02}$ |
| | Lin-AcT | $0.56_{0.17}$ | $0.18_{0.02}$ | $0.36_{0.19}$ | $0.25_{0.02}$ |
| | LinEAS | $\mathbf{0.74}_{0.15}$ | $0.18_{0.02}$ | $\mathbf{0.71}_{0.15}$ | $\mathbf{0.21}_{0.02}$ |
| Tilt-shift | ITI-C | $0.11_{0.11}$ | $0.19_{0.02}$ | $0.44_{0.20}$ | $0.29_{0.01}$ |
| | Lin-AcT | $0.46_{0.20}$ | $0.21_{0.02}$ | $0.48_{0.21}$ | $0.29_{0.01}$ |
| | LinEAS | $\mathbf{0.65}_{0.16}$ | $\mathbf{0.23}_{0.03}$ | $\mathbf{0.65}_{0.15}$ | $\mathbf{0.28}_{0.02}$ |
| Ukiyo-e | ITI-C | $0.20_{0.12}$ | $\mathbf{0.21}_{0.02}$ | $0.47_{0.16}$ | $\mathbf{0.28}_{0.02}$ |
| | Lin-AcT | $0.29_{0.17}$ | $0.20_{0.02}$ | $0.64_{0.15}$ | $0.29_{0.01}$ |
| | LinEAS | $\mathbf{0.48}_{0.18}$ | $0.18_{0.03}$ | $\mathbf{0.73}_{0.12}$ | $\mathbf{0.28}_{0.01}$ |
| Underwater | ITI-C | $0.33_{0.19}$ | $\mathbf{0.21}_{0.02}$ | $0.43_{0.19}$ | $\mathbf{0.26}_{0.01}$ |
| | Lin-AcT | $0.52_{0.21}$ | $0.18_{0.02}$ | $0.41_{0.21}$ | $\mathbf{0.26}_{0.02}$ |
| | LinEAS | $\mathbf{0.59}_{0.21}$ | $0.17_{0.02}$ | $\mathbf{0.56}_{0.20}$ | $\mathbf{0.26}_{0.01}$ |
| Vaporwave | ITI-C | $0.12_{0.17}$ | $\mathbf{0.18}_{0.01}$ | $0.35_{0.17}$ | $\mathbf{0.26}_{0.01}$ |
| | Lin-AcT | $0.33_{0.18}$ | $0.15_{0.02}$ | $0.56_{0.14}$ | $0.27_{0.01}$ |
| | LinEAS | $\mathbf{0.65}_{0.17}$ | $0.16_{0.02}$ | $\mathbf{0.66}_{0.16}$ | $\mathbf{0.26}_{0.02}$ |

Table 15: ImgScore and ClipScore on mitigation for all concepts. Columns 3 and 4 contain results with the original steering direction ($T_\ell(z)$), *i.e.,* removing or mitigating the concepts. Columns 5 and 6 contain results applying the inverse steering operation ($T_\ell^{-1}(z)$), which effectively *induces* the concepts. LinEAS consistently achieves a higher ImgScore for similar ClipScore values.

## O  Image Prompts Dataset

In Section 4.4, we evaluate multiple activation steering methods on DMD2 [30]. To probe different aspects of T2I generation, we query an open-source LLM to generate a new dataset of prompts covering 3 different conditioning categories and 5 concepts per category. We also include a neutral category, which is used as the source in concept addition and target for removal. In Table 16, we show a sample of the dataset, and we include the full dataset in the supplementary material.

| Supercategory | Concept | Example Prompts |
|---|---|---|
| Neutral | | A beaver in its natural habitat. A playful dolphin jumping out of water. A train passing through mountains. A sturdy table with drawers. |
| Style | Vaporwave | Vaporwave neon cityscape at dusk Retro-futuristic arcade machine in a vaporwave setting |
| | Lush | In a lush, overgrown jungle, two young men sitting on a bench and a lady standing next to them. Seagulls in flight with a person feeding one, a lighthouse in the distance, surrounded by a lush, overgrown jungle. |
| | Low-poly | A mountain range at sunrise rendered in low poly style with crisp, angular facets. A futuristic cityscape with neon accents and low poly geometry that creates a digital vibe. |
| | Ukiyo-e | A majestic view of Mount Fuji, cherry blossoms in full bloom, woodblock print style, Ukiyo-e, Edo period aesthetics A stormy sea with giant waves crashing, a lone boat struggling against the current, traditional Japanese woodblock print, Ukiyo-e |
| | Fantasy | A majestic dragon flying over a glowing crystal mountain range, under a purple sky, fantasy art A knight in shining armor standing before a towering, ancient forest, mist swirling around, high fantasy |
| Objects | Robot | A sleek silver robot waves hello in a futuristic city. A tiny robot with glowing eyes explores a dark cave. |
| | Axolotl | A pale pink leucistic axolotl with feathery external gills, smiling serenely in a clear aquarium. A wild-type axolotl, dark and speckled, camouflaged amongst aquatic plants in its natural habitat. |
| | Book | A leather-bound antique book, its gold-leaf title faded, resting on a dusty mahogany desk. A stack of colorful children's picture books, vibrant illustrations peeking from the edges. |
| | Car | A gleaming cherry-red classic 1950s convertible, chrome shining, cruising down a sun-drenched coastal highway, ocean on one side. A rugged, mud-splattered off-road 4x4 vehicle navigating a steep, rocky mountain trail, dust kicking up from its tires. |
| | Hourglass | An antique brass hourglass, its fine golden sand steadily flowing from the top bulb to the bottom, against a dark, moody background. A minimalist, modern glass hourglass with vibrant blue sand, casting a sharp shadow on a white surface. |
| Perspectives | Macro | Macro shot of a ladybug on a vibrant green leaf, its tiny black spots in sharp focus, dewdrop clinging nearby. Extreme macro shot of a honeybee's multifaceted eye, revealing intricate hexagonal patterns. |
| | Fisheye | Fisheye lens perspective of a bustling street market, vibrant stalls and crowds curving dramatically around a central point. Extreme fisheye shot from the center of a packed concert crowd, hands raised, stage lights creating circular flares. |
| | Bokeh | Portrait of a smiling woman, her face in sharp focus, against a background of beautifully blurred city lights creating circular bokeh. A single red rose in perfect focus, its delicate petals detailed, with a creamy green bokeh background of garden foliage. |
| | Underwater | Underwater perspective of a vibrant coral reef teeming with colorful tropical fish, sunlight filtering through clear turquoise water. Sunken pirate shipwreck resting on the sandy ocean floor, schools of fish swimming through its decaying hull, underwater view. |
| | Tilt-shift | Tilt-shift perspective of a bustling city intersection, cars and pedestrians appearing like tiny toys, vibrant colors. A miniature-effect tilt-shift shot of a freight train winding through a verdant, rolling landscape. |

Table 16: Sample of the dataset used for conditioning T2I models. The dataset is divided in 4 different categories: (1) neutral prompts used as source for concept addition and target for removal, (2) style prompts, (3) objects, and (4) perspectives.

