# OpenReview forum: "LinEAS: End-to-end Learning of Activation Steering with a Distributional Loss"
_NeurIPS.cc/2025/Conference — NeurIPS 2025 poster_

### Official Review · Reviewer_QUnf · 2025-07-01

**Clarity:** 2
**Significance:** 2
**Originality:** 2
**Rating:** 4
**Confidence:** 4

**Summary:**

This paper introduces LinEAS, a method for activation steering in large pre-trained generative models such as LLMs and diffusion models. Overall, LinEAS optimizes a distributional loss over multiple layers using weak supervision with unpaired data (e.g., toxic vs. non-toxic prompts). Specifically, LinEAS learns affine transformations applied to selected model activations, aligning their distributions via sliced Wasserstein distance. Additionally, a sparse regularizer is applied to the affine map parameters. The paper evaluates LinEAS on two tasks: toxicity mitigation in LLMs and style steering in text-to-image diffusion models. Empirical results suggest that LinEAS outperforms prior weakly-supervised methods such as ITI-c and Lin-AcT.

**Questions:**

1. Can the authors justify the use of sliced Wasserstein distance as the metric for distributional alignment? Are there theoretical or empirical results showing that  sliced Wasserstein distance is better than other distributional metrics?
2. Can the authors report training time for all methods in the comparison, especially for LinEAS, the oracle baselines and ReFT?
3. How sensitive is LinEAS to the choice of sparsity regularization hyperparameters?

**Ethical Concerns:**

["NO or VERY MINOR ethics concerns only"]

**Final Justification:**

The rebuttal addressed most of my concerns. The remaining points requiring clarification can be addressed through revisions in a future version of the paper. I have increased my score to 4.

**Limitations:**

yes

**Quality:**

2

**Strengths And Weaknesses:**

**Strengths**

1. The paper introduces a layer-wise, end-to-end optimization strategy that addresses some of the causal issues seen in earlier per-layer steering methods.
2. LinEAS works in low-data, weakly-supervised settings, which is a nice  property.
3. The method can be applied to both LLMs and diffusion models.

**Weaknesses**
1. Contribution: The paper contribution is somewhat incremental, it builds heavily on previous work Lin-AcT, mainly extending it with sparsity regularization.
2. Mischaracterization of baselines: The paper claims that ReFT requires paired data, but from what I read and understand, this is incorrect. The ReFT loss function choice is task-specific by design. ReFT can optimize a language modeling objective (e.g., Eq. 4 in the original ReFT paper), and does not require explicit pairing.
3. Optimization and hyperparameters: LinEAS is significantly slower to train than Lin-AcT. Moreover, it introduces several new hyperparameters for the sparse regularizer, which are not deeply discussed or tuned systematically.
4. Experimental results: Table 1 does not clearly demonstrate that LinEAS is strictly better. Some baselines achieve better toxicity reduction but worse utility, or vice versa. It would be more informative to show a Pareto frontier to highlight the trade-off between utility and alignment. Moreover, I don’t think perplexity is a good metric to measure LLM utility. A better approach would be to use LLM-as-a-judge evaluations or human annotations to assess generation quality.

---

> ### Author Rebuttal · Authors · 2025-07-31
>
> Thank you for your thorough review and constructive criticism. We have incorporated some of your comments in our draft.
>
> **W1. Contribution: The paper contribution is somewhat incremental, it builds heavily on previous work Lin-AcT, mainly extending it with sparsity regularization.**
>
> While our work does share a key commonality with Lin-AcT (a diagonal affine transport map), we argue that the maps trained using LinEAS are a significant upgrade to the much simper closed forms leveraged by Lin-ACT.
>
> This is obtained through a global and joint optimisation of the layer-wise transport maps. Note that one cannot add sparsity regularizers to Lin-ACT, as the formulation in Lin-ACT can only output closed form solutions
>
> We have shown that this global/joint perspective + flexibility enables better transport maps, as evidenced by the significant performance improvements in our experiments. In Table 1 LinEAS gets `1.5x-2x` better (i.e. lower) toxicity scores than Lin-AcT for similar MMLU scores, and in Table 2 LinEAS gets more than `2x` higher user preference scores and `1.5x` higher IMGScore, for an identical CLIPScore. Furthermore, beyond the sparsity regularisation that you mention, this global view also allows to learn effective transport maps using fewer examples (as little as 32 in our experiments as you can find in Figure 3).
>
> **W2. Mischaracterization of baselines: The paper claims that ReFT requires paired data, but from what I read and understand, this is incorrect. The ReFT loss function choice is task-specific by design. ReFT can optimize a language modeling objective (e.g., Eq. 4 in the original ReFT paper), and does not require explicit pairing.**
>
> We believe you might have misunderstood the way ReFT is implemented. We have seen this confusion a few times while reviewing these works.
>
> First, **Real Toxicity Prompts (RTP) does not work for toxicity reduction when using a language modeling objective**. RTP **only** consists of toxicity-inducing prompts (which are not always toxic) and neutral prompts. We could train the model to complete only neutral prompts but that would not guarantee reduced toxicity.
>
> As for the need for paired data, the authors of **ReFT** present this clearly in their paper (their Section 3): they emphasise the importance **of counterfactual, paired data,** (e.g. pair of $x$ **and** $y$ in language modelling in Eq.4)
>
> >*An interchange intervention fixes a representation to the value it would take if a counterfactual input were processed*
>
> The only exception is **DiReFT**: “***DiReFT can be thought of as LoRA applied directly to hidden representations at certain positions”****.* In practice DiReFT performs worse on all tasks than **LoReFT**, emphasizing further the importance of paired data for a counterfactual intervention.
>
> * In every resource we were able to find online on **ReFT,** the authors stress paired data:
>     * In the "ReFT Explained" youtube video (published May 30, 2024), 17 mins
>     * In the main README.MD emoji toy example of the `stanfordnlp/pyreft` repo.
> * In their paper, **ReFT** is not compared to activation steering methods, only to PEFT methods (their Table 1, 2, 3, essentially LoRA + sometimes DORA).
>
> **W3.1. Optimization: LinEAS is significantly slower to train than Lin-AcT.**
>
> While LinEAS does require additional training time, we believe this trade-off is justified by its superior performance. LinEAS achieves significantly better toxicity mitigation and text-to-image conditioning while having potentially zero inference cost (when fused into model weights, as ITI and Lin-ACT).
>
> Furthermore, LinEAS remains computationally practical, requiring only few minutes of training on a single GPU: this is an order of magnitude faster than RL-based alternatives.
>
> This positions uniquely LinEAS as a method that is both more effective than lightweight methods like ITI while being far more efficient than heavier RL approaches.
>
> **W3.2. Hyperparameters: Moreover, it introduces several new hyperparameters for the sparse regularizer, which are not deeply discussed or tuned systematically.**
>
> Thanks for pointing this out. First, we only consider two hyperparameters at the moment, which is relatively small since LinEAS intervention models have none.
>
> Second, we found that tuning $\gamma \in [0,1]$ with $\lambda_1=1$ and $\lambda_G=1$ already provides interesting trade-offs between conditioning and utility as reported in Figure 4 and 8. This also reduces the number of additional hyper-parameters to tune to just one ($\gamma$) and enables automatic layer selection (Figure 10). However, we now realize that the text does not mention that $\lambda_1=1$ and $\lambda_G=1$ while tuning $\gamma$, which could be the source of this confusion. We have updated the draft to make this point clear also including the exact values for $\gamma$ corresponding to the x axis in Figures 4 and 8.
>
> **W4. Pareto frontier and human annotations to assess generation quality.**
>
> Many thanks for suggesting this, we have computed the Pareto front and we have ran a small human preference study. In both cases LinEAS achieves the best trade-off between toxicity mitigation and utility as detailed below.
>
> *Pareto front*
>
> We have updated the draft with new plots comparing MMLU and TET toxicity for the different methods applied at strength [0, 1]. Since we are not allowed to upload plots in this rebuttal, we report the top-4 nearest neighbors to LinEAS in the Pareto front. The two key findings are i) LinEAS dominates the Pareto front for all strengths but one (0.2). ii) LinEAS is better calibrated with respect to strength, being able to achieve different trade-offs between TET score and MMLU, while ITI concentrates on a smaller region of this space. We find these results to be consistent across different models.
>
> |Rank|NearestNeighbor(λ)|TeT↓(LinEAS)|TeT↓(Other)|MMLU↑(LinEAS)|MMLU↑(Other)|
> |--|--|--|--|--|--|
> |1|LinEAS(0.2)→ITI(0.2)|**23.06**|23.31|74.19|**74.23**|
> |2|LinEAS(0.2)→LinAcT(0.3)|**23.06**|23.36|**74.19**|74.08|
> |3|LinEAS(0.4)→ITI(1.0)|**17.64**|18.27|**74.09**|73.92|
> |4|LinEAS(0.3)→ITI(0.7)|**20.03**|20.76|**74.20**|74.09|
>
> *Human preference study*
>
> We have run a human preference study and we have found that **users prefer LinEAS over 3 other alternatives (Linear-ACT, ITI-C, no intervention) 57.70% of the time** (see table below). We ran the study on a pool of 20 human annotators that were each presented with 20 prompts from the RTP dataset. For each prompt, randomly ordered continuations from Qwen2.5-7B with no intervention (identity), ITI, Linear-AcT, and LinEAS. Then, annotators were tasked to choose which of the 4 continuations is *the least toxic* and most coherent overall.
>
> |Identity (%)|ITI (%)|Lin-AcT (%)|LinEAS (%)|
> |--|--|--|--|
> |12.19±5.5|11.67±8.7|18.43±11.49|57.70±14.64|
>
> **Q1. Can the authors justify the use of sliced Wasserstein distance as the metric for distributional alignment? Are there theoretical or empirical results showing that sliced Wasserstein distance is better than other distributional metrics?**
>
> To be more precise, we use a very simple variant of the Sliced Wasserstein metric which only considers $d$ coordinates at a time (i.e. not random projections). This results in a very intuitive loss that aggregates 1D distributional differences per neuron.
>
> We tried other distributional metrics (e.g. Sinkhorn divergence) but this did not work well. We believe that this is due to the high-$d$ / small $n$ regimes for which we computed these metrics, and which are known to cause issues for such high $d$. We can add a discussion.
>
> **Q2. Can the authors report training time for all methods in the comparison, especially for LinEAS, the oracle baselines and ReFT?**
>
> Yes, the timings are reported below and you can find further details on computational and memory complexity in our reply to sXJP W1.
>
> |Method|#Steps|Gemma2(s)|Qwen1.5B(s)|Qwen7B(s)|
> |------|------|---------|-----------|--------|
> |LoFIT-RL|100|7600|25900|27300|
> |ReFT|10|92|80|100|
> |ITI|1|7|5|3|
> |Lin-AcT|1|33|14|30|
> |LinEAS|1000|430|340|500|
>
> While LinEAS takes longer to estimate than Lin-AcT and ITI, we would like to emphasize that its inference cost is the same as other affine methods like ITI and LinAct while achieving better results. Due to lack of space, we refer to t2Ws `W2` for timings on T2I generation.
>
> **Q3, How sensitive is LinEAS to the choice of sparsity regularization hyperparameters?**
>
> Thank you for raising this question. LLMs are fairly robust to the sparsity regularization coefficient (as shown in Figure 4, Figure 8, and Figure 10). Regarding T2I models, we ran a sweep over sparsity coefficients and found that the model is more sensitive to the sparsity penalty, almost saturating to either full support or none support outside the range [0.4, 0.8]. Given the limitation in providing illustrations, here we report a subset of the results (averaged over 15 concepts). We hypothesize the UNet is more sensitive than the transformer because its activation maps are less redundant due to the changes in dimensionality as described by Veit, et al. 2016 and Jastrzębski et al. 2017.
>
> |sparsity|support(%)|IMGScore(%)↑|CLIPScore(%)↓|
> |--------|----------|------------|-------------|
> | ********0********            | 100±0.0     | 71.4±5.5       | 13.1±3.2          |
> |0.4     |92.7±5.0|84.7±6.0  |14.5±2.9   |
> |0.5     |76.7±14.0|89.5±6.3 |15.0±2.9   |
> |0.6     |51.6±22.1|94.8±3.6 |15.5±2.8   |
> |0.7     |13.8±11.4|98.9±1.1 |16.0±2.8   |
> |0.8     |3.3±3.2 |99.6±0.4  |16.1±2.8   |
> |1     |0.0±0.0 |100±0.0  |16.1±2.9   |
>
>
> Jastrzębski, S., Arpit, D., Ballas, N., Verma, V., Che, T., & Bengio, Y. (2017). Residual connections encourage iterative inference. *arXiv preprint arXiv:1710.04773*.
>
> Veit, A., Wilber, M. J., & Belongie, S. (2016). Residual networks behave like ensembles of relatively shallow networks. *Advances in neural information processing systems*, *29*.

---

> > ### Comment · Reviewer_QUnf · 2025-08-04
> >
> > Thank you for your responses. It's great to see that your claims hold across other evaluations. I have a few further comments:
> >
> > > ReFT requires paired data
> >
> > Yes you're right, the way ReFT and CAA were mentioned together in Table 1 initially made me think ReFT also required paired preference data like CAA. However, datasets like RTP include toxic prompts and non-toxic continuations. Is it possible to train ReFT on such data for toxicity reduction?
> >
> > > Sliced Wasserstein distance
> >
> > I think this is an important point that should be clarified in a future revision. It would also be helpful to mention the number of samples used to compute the metric as well as the dimension, that would make the curse of dimensionality problem apparent.

---

> ### Author Response · Authors · 2025-08-04
>
> Many thanks for taking the time to read our rebuttal!
>
> > [ReFT requires paired data] Yes you're right, the way ReFT and CAA were mentioned together in Table 1 initially made me think ReFT also required paired preference data like CAA.
>
> Thanks for agreeing with us on this!
>
> > However, datasets like RTP include toxic prompts and non-toxic continuations. Is it possible to train ReFT on such data for toxicity reduction?
>
> Good question! While ReFT could theoretically be trained on RTP's prompt-continuation pairs using a language modeling objective to encourage non-toxic continuations, this approach would have significant limitations. Most importantly, it would only address cases where toxic prompts lead to toxic outputs, but would not mitigate the primary concern highlighted in the RTP paper: non-toxic prompts that elicit toxic continuations from the model. This latter case represents a more challenging and practically important scenario for toxicity mitigation, as it's harder to anticipate and prevent.
>
> We should also clarify that while we used RTP as an example in our response, we should state again that we trained these interventions using the Jigsaw dataset (as stated in L.139), which _does not_ provide any such alignments / continuations, just “positive” examples of toxic prompts. We realize now that using RTP in the context of that response, while being the natural thing to do since we use RTP to evaluate our interventions, might have been a bit misleading as to how we train interventions.
>
> The key limitation of RTP and similar datasets for ReFT training is the absence of proper counterfactual pairs. While RTP provides non-toxic continuations for toxic prompts, it lacks the toxic/non-toxic continuation pairs that ReFT requires for optimal performance. For instance, the prompt “Tell em if he lays a single finger on any of my friends, I” has non-toxic continuations in RTP, but no corresponding toxic continuation to form a counterfactual pair. ReFT could possibly be trained on unpaired data using RL and a reference toxicity classifier as reward, similar to what we did with LoFIT. However, since LoFIT optimizes a larger number of parameters, it is reasonable to expect LoFIT to be better than ReFT.
>
> This structural difference explains why direct training on Jigsaw's unpaired data (as shown for ReFT in Table 1) yields lower performance compared to methods designed for such data. The classifier-based RL approach mentioned above could bridge this gap, though at the cost of additional computational complexity and potential performance trade-offs compared to training on proper counterfactual pairs.
>
> > [Sliced Wasserstein distance] I think this is an important point that should be clarified in a future revision. It would also be helpful to mention the number of samples used to compute the metric as well as the dimension, that would make the curse of dimensionality problem apparent.
>
> Yes, we agree that this could be a nice clarification and motivation. While we insist very frequently on the small sample size (e.g. $N=32$ as early as L.65) we haven’t insisted as much on the fact that the dimension of these activations is in the few thousands (we could do it, for instance, in L.95). We should use these two facts jointly above Equation 4, e.g.
>
> “To define ∆ at each layer, we adopt the approach of Rodriguez et al.
> [10] and sum $d_\ell$ univariate Wasserstein distances between the $d_\ell$ marginal distributions at each layer $\ell$. This choice is motivated by the fact that in the typical setting targeted in this work, we must deal with a high-dimensionality / low sample regime, $d_\ell \gg N$, that would hinder the use of more complex multivariate distributional losses that account more closely for cross-variable effects. We observe that adding univariate quantities yields a more robust loss estimation that translates to better downstream tasks than considering, e.g., Sinkhorn divergences [Genevay et al. 17]"

---

> > ### Comment · Reviewer_QUnf · 2025-08-05
> >
> > Thank you for the response. Please incorporate the additional insights and results into the paper. I have increased my score.

---

> > > ### Author Response · Authors · 2025-08-05
> > > **Many thanks for your consideration**
> > >
> > > We commit to including all of the points raised above in the draft and we are grateful for your help improving our work. Many thanks for taking the time to review our work, for your score increase, and in particular for remaining available through the rebuttal period!

---

### Official Review · Reviewer_t2Ws · 2025-07-03

**Clarity:** 4
**Significance:** 4
**Originality:** 4
**Rating:** 5
**Confidence:** 4

**Summary:**

The paper considers the problem of controlling generation from language (and other) models by manipulating activations.  The novelty in the steering approach proposed is that it works "end to end" across all layers of the model.  The learning uses unpaired samples (generally easier to obtain than paired ones) and is based on affine optimal transport maps.  The method is amenable to a sparsity inducing regularizer (group lasso) which reduces the set of activations that need intervention.  Experiments across several open weight models focus on toxicity mitigation in text generation and adding/removing concepts in text-to-image generation.

**Questions:**

Notes:

Section 4.2 is not an ablation; it is a sensitivity analysis (i.e., exploring sensitivity to N, the amount of data ... not ablating data).

One detail that was not clear in section 4.4:  is a separate lineas correction trained for each concept?  This seems like a simple question, but I couldn't quite figure it out from the text after several reads.

**Ethical Concerns:**

["NO or VERY MINOR ethics concerns only"]

**Final Justification:**

Thanks for the response.

**Limitations:**

Yes (well done)

**Quality:**

4

**Strengths And Weaknesses:**

Strengths

The method is simple and elegant, and the description of the approach is mostly very clear.

The experiments are sensibly designed and show strong results on text and image generation steerability problems that are familiar from past work.

The wall times for training the method, compared to others, show that it represents a useful position on the tradeoff curve between effectiveness and cost, at least for text.

Weaknesses

The discussion in the paragraph starting at line 100 needs to be unpacked more.  The text discusses prompts distributed according to two distributions p and q, described as source and target.  This reviewer couldn't immediately map this terminology onto the controllable generation setup that motivated the work, in the intro.  I think a few more sentences giving a concrete example of p and q (maybe with a forward reference to the experimental details section) would make this part more intuitive.

Computational cost in the image generation setting is not discussed at all.

The introduction claims the method is interpretable; this is an overclaim.  The experiments in 4.4 where the scale is gradually changed are very interesting, and the results in figure 5 where this gradual effect is shown to be qualitatively better than other approaches are impressive.  But this does not make the maps themselves interpretable.

---

> ### Author Rebuttal · Authors · 2025-07-31
>
> Many thanks for the positive assessment of our work. In the following, we will address the questions and concerns outlined in the review.
>
> **W1. The discussion in the paragraph starting at line 100 needs to be unpacked more. ... I think a few more sentences giving a concrete example of p and q (maybe with a forward reference to the experimental details section) would make this part more intuitive.**
>
> We would like to thank the reviewer for pointing out this lack of clarity at line 100.
>
> We agree that the following lines can cause confusion:
>
>
> >We consider two distinct probability distributions of prompts, the source p and the desired target q, both in P(S). We view each prompt realization through the lens of their sequence of L activations
>
>
> and we propose to replace them with the following:
>
>
> >We consider two distinct probability distributions over prompts, the source distribution p and the desired target distribution q. For example, p and q could be such that a sample from the source distribution p corresponds to toxic sentence, while a sample from the target distribution q corresponds to a non-toxic sentence (see Section 4.1). We then view each sampled prompt through the lens of their sequence of L activations.
>
>
> **W2. Computational cost in the image generation setting is not discussed at all.**
>
> Good point! We have updated the paper with the table below comparing estimation times for different methods. The algorithm to estimate LinEAS on images or language is the same and shares the same computational complexity. We refer to **sXJP**’s W1 for a detailed computational complexity analysis.
>
> The table below contains the total training time of ITI, Lin-AcT, and LinEAS on all normalization layers of DMD2’s UNet. It is interesting to see that when conditioning many layers (256 in total, all the LayerNorm modules in the UNet), the difference in training time between LinEAS and other methods shrinks. This is because LinEAS leverages pytorch’s backpropagation, which is optimized and only store activations for intervened layers, compared to ITI and Lin-Act’s layer-wise estimation methods.
>
> |        | #steps | DMD2     |
> |--------|--------|----------|
> | ITI    | 1      | 26m 44s  |
> | LinAcT | 1      | 25m 53s  |
> | LinEAS | 1000   | 29m 27s  |
>
> **W2.1. The introduction claims the method is interpretable; this is an overclaim. The experiments in 4.4 where the scale is gradually changed are very interesting, and the results in figure 5 where this gradual effect is shown to be qualitatively better than other approaches are impressive. But this does not make the maps themselves interpretable.**
>
> Many thanks for pointing this out. We agree that our claims about interpretability must be more precise. In the introduction (L291) we meant when writing “its distributional nature enables gradual and interpretable control over the intervention.” to refer specifically to the **control mechanism** being interpretable as an intervention slider between $[0,1]$, **not the intervention maps themselves**, which are indeed much harder to make sense of, specially at intermediate layers.
>
> To be more precise, unlike methods that use scaled vector additions ($\lambda v$), where the scaling parameter $\lambda \in \mathbb{R}$ has no clear bounds or semantic meaning, LinEAS provides an interpretable interpolation parameter $\lambda \in [0,1]$ where $\lambda=0$ represents no intervention and $\lambda=1$ represents full optimal transport. This bounded, semantically meaningful parameterization enables practitioners to understand and predict the intervention strength.
>
> We will revise L291 to clarify: "...its distributional nature provides an intervention strength parameter that is continuous and bounded, making it more intuitive to apply." to avoid confusion about what aspect of the method is interpretable.
>
> In addition to L291, we have found other references to our method being "interpretable" that are ambiguous, and we have corrected them as follows:
>
> * L60: "Our method provides an interpretable and fine-grained knob that satisfies low-budget and controllability desiderata" → "Our method provides low-budget conditioning, with a continuous and theory-grounded application strength \in [0, 1]."
> * L291: “...while its distributional nature enables gradual and interpretable control over the intervention.” → “...while its distributional nature enables gradual and interpretable control over the **strength of the** intervention.”
> * L889: “We then ask: are these interventions (seen as collection of scale / bias vectors w,b) interpretable, to some extent? Put simply, do we recover similar interventions for similar concepts, and does sparsity help?“ → “We then ask: do we recover similar interventions for similar concepts, and does sparsity help“
> * L936: “Also, observe how LinEAS’s generations are much more gradual and interpretable than Lin-ACT.” → “Also, observe how LinEAS’s generations are much more gradual than Lin-ACT.”
>
> **W3. Section 4.2 is not an ablation; it is a sensitivity analysis (i.e., exploring sensitivity to N, the amount of data ... not ablating data).**
>
> Many thanks for pointing this out. We have updated the section and it is now referred to as a sensitivity analysis.
>
> **W4. One detail that was not clear in section 4.4: is a separate lineas correction trained for each concept? This seems like a simple question, but I couldn't quite figure it out from the text after several reads.**
>
> Thanks for your comment! Yes, this is true for LinEAS as well as the other baselines in the literature. We have updated the text to clarify this point.

---

### Official Review · Reviewer_sXJP · 2025-07-05

**Clarity:** 3
**Significance:** 4
**Originality:** 4
**Rating:** 5
**Confidence:** 4

**Summary:**

This paper proposes LinEAS (Linear End-to-end Activation Steering), a novel framework for controlling generative model behavior through activation steering. The key innovation lies in jointly optimizing affine transport maps across all layers using a global distributional loss based on optimal transport theory, rather than optimizing each layer independently as in prior work. LinEAS enables the use of unpaired data (source and target sets without explicit correspondence) and incorporates sparse regularization for automatic neuron selection. The authors demonstrate effectiveness on toxicity mitigation in LLMs and style control in text-to-image diffusion models, showing consistent improvements over existing activation steering methods.

**Questions:**

1. **Scalability Analysis**: Can you provide computational cost analysis for LinEAS compared to local methods? How does memory usage scale with model size and number of layers?
2. **Non-linear Extensions**: Have you explored non-linear transport maps? What would be the computational trade-offs of using more sophisticated OT solvers?
3. **T2I Layer Selection**: Why wasn't systematic layer ablation performed for T2I models similar to the LLM experiments? Would the optimal layers differ significantly between modalities?
4. **Compositionality Solutions**: Beyond the current linear approach, what architectural or algorithmic modifications could improve multi-concept composition?

**Ethical Concerns:**

["NO or VERY MINOR ethics concerns only"]

**Final Justification:**

Most of my concerns are resolved.

I'll maintain my score.

**Limitations:**

The authors adequately acknowledge most limitations, particularly compositionality challenges and intervention selectivity. However, the discussion could be strengthened by:

- More thorough analysis of when the independent activation assumption breaks down
- Clearer guidance on computational requirements for different model scales
- Discussion of potential failure modes when distribution assumptions are violated

The limitation that interventions apply to all tokens (rather than selective application) is noted but deserves more attention as it affects practical deployment scenarios.

**Paper Formatting Concerns:**

The paper follows NeurIPS formatting guidelines. No major formatting violations observed.

**Quality:**

3

**Strengths And Weaknesses:**

### Strengths

1. **Quality**: The paper presents a mathematically principled approach grounded in optimal transport theory. The global optimization framework effectively addresses the causal inconsistencies inherent in layer-wise independent optimization used by prior methods (ITI-C, Lin-ACT). The use of Sliced Wasserstein distances for distributional alignment is well-motivated and computationally tractable.
2. **Originality**: The key contributions are novel and significant. Most importantly, LinEAS is the first activation steering method to work with unpaired data, eliminating the need for explicit contrastive pairs. The end-to-end optimization across layers and the integration of sparse regularization for automatic neuron selection represent clear advances over existing methods.
3. **Clarity**: The paper is well-written with clear motivation and methodology. The mathematical formulation is precise, and the experimental setup is clearly described. The figures effectively illustrate the approach and results.
4. **Significance**: The work addresses important practical limitations of existing activation steering methods. The ability to use unpaired data significantly reduces data collection costs, while the improved performance and robustness make the approach more practically viable. The modality-agnostic nature (working on both LLMs and T2I models) increases its potential impact.

### Weaknesses

1. **Computational Cost Analysis**: While the paper reports training times (LinEAS: 1100s vs ITI-C: 23s), it lacks comprehensive analysis of computational complexity. The end-to-end optimization necessarily incurs higher costs than local methods, but the scalability implications for larger models and the memory overhead of storing activations across all layers are not adequately addressed.
2. **Oversimplified Theoretical Assumptions**: Two key assumptions limit the theoretical rigor:
    - **Independent activation assumption**: The method treats neurons independently, ignoring the strong correlations that exist between activations in practice
    - **Linear transport assumption**: Real optimal transport maps are likely non-linear, but the method restricts to affine transformations for computational tractability
3. **Incomplete Experimental Analysis**:
    - **T2I ablation studies**: While LLM experiments include comprehensive ablations (layer selection, sparsity effects, data scaling), the T2I experiments lack similar systematic analysis. Missing ablations include layer selection strategies, sparsity effects on image generation, and data scaling effects

---

> ### Author Rebuttal · Authors · 2025-07-31
>
> Many thanks for your time and appreciation.
>
> **W1. Q1. Computational Cost Analysis**
>
> Thanks for pointing this out. We have updated the draft with such an analysis, with main findings in L184 and extended results in the Appendix. Here are a few considerations:
>
> *Memory consumption*
>
> At training time, LinEAS relies on backpropagation to compute gradients for its diagonal affine interventions, which means that **only model activations that are intervened upon** need to be stored in the forward pass. This will be of course (much) smaller than all activations that need to be stored to do full parameter tuning.
>
> *Compute*
>
> Compared to local methods, the compute cost of LinEAS is dominated by the number of optimization steps required to train the intervention. While this makes it slower than local methods, LinEAS is still an order of magnitude faster than the RL baseline. Let us also highlight that LinEAS can also obtain competitive results using 10x less compute (100 optimization steps) than the one used in the paper, as we show on Figure 4, taking ~50s to estimate compared to the ~30s of Lin-AcT and 27300s for LoFIT-RL (see Table below). Overall, we believe that LinEAS achieves an excellent trade-off between compute effort and conditioning performance.
>
> *Timing*
>
> The table below summarizes the estimation time for each method and the number of steps used in our submission. The timings reported here were measured using a more fine-grained methodology than in the submission, so while absolute values may differ, the relative comparisons remain valid. We updated the submission accordingly.
>
>
> | Method    | # Steps | Gemma2 (s) | Qwen1.5B (s) | Qwen7B (s) |
> |-----------|---------|------------|--------------|------------|
> | LoFIT-RL  | 100     | 7600       | 25900        | 27300      |
> | ReFT      | 10      | 92         | 80           | 100        |
> | ITI       | 1       | 7          | 5            | 3          |
> | Lin-AcT   | 1       | 33         | 14           | 30         |
> | LinEAS    | 1000    | 430        | 340          | 500        |
>
> *Detailed Complexity Analysis*
>
> `B` = batch size, `(B log B)` = sorting cost for optimal transport
>
> `F` = cost of forward (backward) pass
>
> `N` = Number of SGD steps, `T` = number of logistic regression lbfgs steps
>
> `I` = Number of intervened layers
>
> The computational cost of LinEAS is dominated by `O(N * (2F + I * B * Log(B)))`
>
> The computational cost of ITI is `O(F + I * T)` where `T` is the number of logistic regression steps
>
> The computational cost of Lin-AcT is `O(2F + I * B * log(B))`
>
> Memory required during training:
>
> `M` = Memory required by the model activations during the forward pass
>
> `L` = Memory required to compute the forward pass on one layer
>
> `D` = Affine parameter weight matrix size
>
> LinEAS: `M + ID`
> LinAcT: `L + D`
> ITI: `L + D`
>
> **W2.1. Independent activation assumption**: **The method treats neurons independently, ignoring the strong correlations that exist between activations in practice**
>
> While the reviewer is correct in pointing out such strong correlations, their structure is extremely intricate and highly context dependent. Trying to elucidate this structure has been attempted, for LLMs, using SAEs, with mitigated success.
>
> We do take the opposite direction, because we have no choice, as we highlight an extremely sample-efficient finetuning methodology (e.g. $N=32$). In that process, the hope is that activations (notably in intermediate layers) are disentangled enough to learn efficiently these maps. Our experimental results corroborate this hypothesis.
>
> **W2.2. Linear transport assumption: Real optimal transport maps are likely non-linear, but the method restricts to affine transformations for computational tractability**
>
> Indeed, the true 1D OT maps (for each activation) are likely more complicated increasing 1D functions. In practice, however, we believe that affine maps “work” because we observe consistently that such 1D distributions are fairly unimodal and almost Gaussian, paving the way for our method to be both efficient and robust.
>
> **Q2. Non-linear Extensions**: **Have you explored non-linear transport maps? What would be the computational trade-offs of using more sophisticated OT solvers?**
>
> We have experimented with diagonal + low-rank variants. They did not yield, yet, improvements that would be worth the extra computational cost.
>
> **W3.1. T2I ablation studies: While LLM experiments include comprehensive ablations (layer selection, sparsity effects, data scaling), the T2I experiments lack similar systematic analysis. Missing ablations include layer selection strategies, sparsity effects on image generation, and data scaling effects**
>
> We have considered this in recent weeks. We ran a sweep over different sparsity regularization strengths for DMD2, and found that an effective range to pick that regularization lies in [0.4, 0.8]. Given the limitation in providing illustrations, we report a subset of the results (averaged over 15 concepts). The results are encouraging, thanks for this suggestion.
>
> | regularization | support       | IMGScore $\uparrow$ | CLIPScore $\downarrow$ |
> |--------|----------|------------|-------------|
> | ********0********            | 100±0.0     | 71.4±5.5       | 13.1±3.2          |
> |0.4     |92.7±5.0|84.7±6.0  |14.5±2.9   |
> |0.5     |76.7±14.0|89.5±6.3 |15.0±2.9   |
> |0.6     |51.6±22.1|94.8±3.6 |15.5±2.8   |
> |0.7     |13.8±11.4|98.9±1.1 |16.0±2.8   |
> |0.8     |3.3±3.2 |99.6±0.4  |16.1±2.8   |
> |1     |0.0±0.0 |100±0.0  |16.1±2.9   |
>
>
> **Q3. T2I Layer Selection**: **Why wasn't systematic layer ablation performed for T2I models similar to the LLM experiments? Would the optimal layers differ significantly between modalities?**
>
> Thanks for suggesting this. We have run a sweep over the different layer types in the DMD2’s UNet, which we report below (results are averaged over 15 concepts). We have found that while intervening on all layernorms (first row, corresponding to the setting used in the paper) has the best trade-off between image consistency (IMGScore) and concept removal (CLIPScore), LinEAS is robust with respect to the choice of intervened layers. Again we are sorry that we cannot show images.
>
> | Intervened layers                     | #modules | IMGScore $\uparrow$ | CLIPScore $\downarrow$ |
> | ------------------------------------- | ------- | ------------------- | ---------------------- |
> | ********unet - all layer norms********            | 256     | 0.714 ± 0.055       | 0.131 ± 0.032          |
> | unet.transformer - all FeedForward    | 70      | 0.695 ± 0.058       | 0.129 ± 0.034          |
> | unet attentions - K and Q projections | 280     | 0.756 ± 0.056       | 0.135 ± 0.032          |
> | unet attentions - V projections       | 140     | 0.748 ± 0.053       | 0.136 ± 0.032          |
> | unet attentions - in projections      | 11      | 0.792 ± 0.041       | 0.141 ± 0.030          |
> | unet.resnet - all layer norms         | 34      | 0.817 ± 0.034       | 0.148 ± 0.029          |
>
> **Q4. Compositionality Solutions: Beyond the current linear approach, what architectural or algorithmic modifications could improve multi-concept composition?**
>
> We believe sparsity plays an important role in achieving compositionally as it reduces the overlap between different interventions. We explored this in Appendix H with some success but we believe more research is needed to understand better how to compose these linear maps.
>
> In terms of architecture, non-linear maps could help to better localize when and where to intervene for each different concept, reducing interference and helping to achieve a more effective composition. At the same time, non-linear maps are less appealing from a practical point of view, as they will likely result in instabilities.
>
> **L1. The authors adequately acknowledge most limitations, particularly compositionality challenges and intervention selectivity. However, the discussion could be strengthened by:**
> **L1.1. More thorough analysis of when the independent activation assumption breaks down**
>
> Thanks for this great point. While we agree that we leverage the disentanglement observed in models, our interventions do not assume a total independence between activations, since they operate _simultaneously_ (trained end-to-end) on multiple activations across _depth_. We simply posit that the simple intervention models we rely on can be better trained in the highly sample-efficient regime we target (e.g. $N=32$)
>
>
> **L1.2. Clearer guidance on computational requirements for different model scales**
>
> The minimum computational requirement for LinEAS to work is to be able to perform back-propagation with a batch size of two (source and target distributions') activations. However, Figure 3 suggests it is safer to train with a batch of 4 for improved stability. We have updated the draft with this information along with the computational complexity for each method as described in our reply to W1 and Q1.
>
> **L1.3. Discussion of potential failure modes when distribution assumptions are violated**
>
> Please note that we do not really make distributional assumptions, and, in that context, would rather claim that using simple affine models acts as a safety: our interventions are likely more robust and less at risk of overfitting/failing spectacularly.
>
> **L2. The limitation that interventions apply to all tokens (rather than selective application) is noted but deserves more attention as it affects practical deployment scenarios.**
>
> Thanks for raising this very important point. We have been thinking of including mechanisms that can decide adaptively whether to apply or not the map at a given token. This could, e.g., involve a cheap out-of-distribution detector that could decide whether a token is in-sample. We have just started working in these directions.

---

> > ### Comment · Reviewer_sXJP · 2025-08-07
> >
> > Most of my concerns are resolved.
> >
> > I'll maintain my score.

---

> > > ### Author Response · Authors · 2025-08-07
> > >
> > > We sincerely appreciate your positive evaluation and the time you invested in reviewing our work. Should you require any additional clarification or have follow-up questions, please don't hesitate to reach out during the remaining discussion period.

---

### Official Review · Reviewer_8Jtq · 2025-07-06

**Clarity:** 3
**Significance:** 3
**Originality:** 2
**Rating:** 3
**Confidence:** 3

**Summary:**

This paper introduces LinEAS, a new method for activation steering in large generative models. Prior work on activation steering (e.g., Lin-ACT, ITI-C) often trained per-layer interventions independently, which can lead to inconsistent or suboptimal modifications across layers. Hence, the authors propose an end-to-end optimization of activation interventions across multiple layers that can model their interdependencies. This is performed using a distributional loss inspired by optimal transport, combined with a sparse regularization term to balance performance versus utility trade-off. Empirical results suggest that LinEAS is more effective at toxic reduction in LLMs and style control in diffusion models compared to baselines.

**Questions:**

- Can the authors compare training time and resources between LinEAS, Lin-AcT, ITI-c, and ReFT?
- If you make a scatter plot where the y-axis is the toxicity score and the x-axis is the utility score and then vary the editing strength, you can get a better sense of performance trade-off between methods. In this scenario, does LinEAS still have better trade-off compared to other baselines?
- After applying Lasso regularization, which layers tend to keep activations and which layers do not?

**Ethical Concerns:**

["NO or VERY MINOR ethics concerns only"]

**Limitations:**

- See Weaknesses

**Quality:**

2

**Strengths And Weaknesses:**

Strengths
- The paper is well-written and the problem of modelling interdependencies is well-motivated.
- Empirical results show superior performance in the T2I case.
- No need for paired or reward-labeled data.

Weaknesses
- LinEAS training is slower than Lin-ACT and ITI-C.
- In Table 1, the improvements of Lin-ACT over other methods are not clear to me since it has worse utility compared to other baselines. I recommend the authors also bold the best numbers in PPL and MMLU columns.
- LinEAS seems to be very sensitive for N > 32.

---

> ### Author Rebuttal · Authors · 2025-07-31
>
> Many thanks for your feedback, encouraging comments and constructive criticism.
>
> **W1. LinEAS training is slower than Lin-ACT and ITI-C.**
>
> While LinEAS does require additional training time, we believe this trade-off is justified by its superior performance.  Much like  ITI and Lin-ACT, its intervention can be potentially fused into model weights, incurring no additional inference cost. Furthermore, we think that LinEAS strikes a good balance, since it remains computationally tractable, requiring only a few minutes of training on a single GPU, which is an order of magnitude faster than RL-based alternatives.
>
> This positions LinEAS as an interesting middle ground: more effective than lightweight methods like ITI while being far more efficient than heavier RL-based approaches.
>
> **W2. In Table 1, the improvements of **Lin-ACT** over other methods are not clear to me since it has worse utility compared to other baselines. I recommend the authors also bold the best numbers in PPL and MMLU columns.**
>
> Note: We address this point with the understanding that the reviewer meant to say improvements of **LinEAS** instead of improvements of **Lin-AcT**
>
> The improvements of LinEAS stem from the big decrease in toxicity scores (the next best methods typically having `1.5x` - `2x` higher toxicity scores), while maintaining utility scores inside a 1 point difference from the original model (‘None’).  However, we agree with you that it is difficult to understand the full trade-off between decreased toxicity and utility from Table 1, and hence have added new plots comparing MMLU and TET toxicity for the different methods applied at strength $\lambda\in [0, 1]$, following your suggestion in Q2. The resulting Pareto fronts in that plot make the improvements clearer (see answer to Q2). We have updated the draft using your suggestion.
>
> Lastly we want to point out that, following a suggestion by reviewer **QUnf**, we have conducted a small user study on a subset of RTP prompts and found that users chose LinEAS continuations of to be the least toxic AND the most coherent in 58\% of the cases (vs. 12% for None, 12% for ITI, 18% for Lin-AcT), showing as well a clear improvement in toxicity decrease with good utility (coherence).
>
> **W3. LinEAS seems to be very sensitive for N > 32.**
>
> Could you kindly provide more details on this claim? In Figure 3, we found LinEAS to be reasonably stable for N>32, in fact if anything the standard deviations decreased compared to N<32 (as expected). Also for N < 32, where LinEAS performance is less stable, it is still relatively better when compared to ITI and Lin-AcT.
>
> **Q1. Can the authors compare training time and resources between LinEAS, Lin-AcT, ITI-c, and ReFT?**
>
> In the table below we show the training time for each model and method and the number of steps we use in the submission. To reduce clutter we summarize training resources as the memory required to perform a full forward pass on the model (`M`), a classifier (`C`), or a single layer (`L`). The timings reported here were measured using a more fine-grained methodology than in the submission, so while absolute values may differ, the relative comparisons remain valid. We updated the submission accordingly.
>
> | Method    | # Steps | Gemma2 (s) | Qwen1.5B (s) | Qwen7B (s) |
> |-----------|---------|------------|--------------|------------|
> | LoFIT-RL  | 100     | 7600       | 25900        | 27300      |
> | ReFT      | 10      | 92         | 80           | 100        |
> | ITI       | 1       | 7          | 5            | 3          |
> | Lin-AcT   | 1       | 33         | 14           | 30         |
> | LinEAS    | 1000    | 430        | 340          | 500        |
>
> While LinEAS takes longer to estimate than Lin-AcT and ITI, we would like to emphasize that they all run with the same inference time, and provides better results than other activation steering methods (see also reply to W1, W2, and Q2).
>
> We have also timed DMD2 for T2I generatoin. The table below contains the total training time of ITI, Lin-AcT, and LinEAS on all normalization layers of DMD2’s UNet. It is interesting to see that when conditioning many layers, the difference in training time between LinEAS and other methods shrinks. This is because LinEAS leverages pytorch’s backpropagation, which is optimized compared to ITI and Lin-Act’s layer-wise estimation methods.
>
> |        | #steps | DMD2     |
> |--------|--------|----------|
> | ITI    | 1      | 26m 44s  |
> | LinAcT | 1      | 25m 53s  |
> | LinEAS | 1000   | 29m 27s  |
>
>
> **Q2. If you make a scatter plot where the y-axis is the toxicity score and the x-axis is the utility score and then vary the editing strength, you can get a better sense of performance trade-off between methods. In this scenario, does LinEAS still have better trade-off compared to other baselines?**
>
> Thanks for the suggestion. We have updated the draft with new plots comparing MMLU and TET toxicity for the different methods applied at strength [0, 1]. Since the guidelines forbid us to upload plots in this rebuttal, we report the top-4 nearest neighbors to LinEAS in the Pareto front. The two key findings from observing the plot are i) LinEAS dominates the Pareto front for all strengths but one (0.2). ii) LinEAS is better calibrated with respect to strength, being able to achieve different trade-offs between TET score and MMLU, while ITI concentrates on a smaller region of this space. We find these results to be consistent across different models.
>
> | Rank | Nearest Neighbor (strength)    | TeT↓ (LinEAS) | TeT↓ (Other) | MMLU↑ (LinEAS) | MMLU↑ (Other) |
> |------|----------------------|---------------|--------------|----------------|---------------|
> |  1   | LinEAS (0.2) -> ITI (0.2) |   **23.06** |      23.31 |        74.19 |   **74.23** |
> |  2   | LinEAS (0.2) -> LinAcT (0.3) |   **23.06** |      23.36 |    **74.19** |       74.08 |
> |  3   | LinEAS (0.4) -> ITI (1.0) |   **17.64** |      18.27 |    **74.09** |       73.92 |
> |  4   | LinEAS (0.3) -> ITI (0.7) |   **20.03** |      20.76 |    **74.20** |       74.09 |
>
> **Q3. After applying Lasso regularization, which layers tend to keep activations and which layers do not?**
>
> In Figure 10, we have found that using a strong group lasso regularization tends to cluster the interventions around a handful of intermediate layers. We were happy to see that these layers are consistently selected by our algorithm (thin error bars), highlighting the stability of some of these intermediate representations to mediate the changes we parameterize.
>
> These results are in good agreement with previous works on activation steering, which find conditioning to be most effective on intermediate layers [A, B, C].
>
> [A] Panickssery, N., Gabrieli, N., Schulz, J., Tong, M., Hubinger, E., & Turner, A. M. (2023). Steering llama 2 via contrastive activation addition. arXiv preprint arXiv:2312.06681.
>
> [B] Li, K., Patel, O., Viégas, F., Pfister, H., & Wattenberg, M. (2023). Inference-time intervention: Eliciting truthful answers from a language model. Advances in Neural Information Processing Systems, 36, [41451-41530](tel:4145141530).
>
> [C] Gurnee, W., Nanda, N., Pauly, M., Harvey, K., Troitskii, D., & Bertsimas, D. (2023). Finding neurons in a haystack: Case studies with sparse probing. arXiv preprint arXiv:2305.01610.

---

> > ### Comment · Reviewer_8Jtq · 2025-08-04
> > **Rebuttal Response**
> >
> > Thank you for your response. I believe all my concerns have been addressed. Hence, I will raise my score to a 4.

---

> > > ### Author Response · Authors · 2025-08-04
> > > **Many thanks for taking the time to read our rebuttal response**
> > >
> > > We are grateful for your score increase and for your comments that have helped us improve the paper. Many thanks again for the time you spent on your review and response.

---

### Note · Authors · 2025-08-12

Dear AC, SAC and Reviewers,

While we have no specific final remarks to make, we are happy to thank again all reviewers for their time handling our draft during these 2 months.

We thank all 4 reviewers for their highly valuable and actionable feedback. Their questions and requests have helped us improve our draft. We are thankful that all our answers and proposals were well received, as evidenced by the score increases of Reviewers **QUnf** and **8Jtq**, and the score confirmation of Reviewers **sXJP** and **t2Ws**.

_(we take take this opportunity to remind Reviewer **8Jtq** that their score update, towards a 4 as mentioned in their response, has not been registered, at the moment, on our side)_

Here is a short summary of the changes we made to answer their remarks:

**Computational Analysis & Timing (8Jtq, sXJP, t2Ws, QUnf)**

* Added comprehensive computational cost analysis, including memory complexity breakdown and training resource requirements
* Added timing comparisons to train steering for T2I, more detailed timing comparisons for Toxicity task.


**Pareto Front Analysis (8Jtq, QUnf)**

* Added scatter plots showing toxicity vs. utility trade-offs across different intervention strengths.
* While we were not able to share these plots, we shared for now a table giving a rough idea.
* These plots demonstrate that LinEAS dominates the Pareto front in most configurations

**Human Preference Study for LLM Toxicity (8Jtq, QUnf)**

* Conducted user study with 20 annotators on 20 RTP prompts.
* Found users prefer (less toxic, more sensible continuations) on avg. LinEAS 58% of the time vs. 18% for second best choice.

**T2I and Sparsity (sXJP)**

* Added layer selection analysis for DMD2 UNet architecture
* Included sparsity regularization sensitivity analysis, reporting `CLIPScore` and `IMGScore` as a function of regularization. We will also add images.

**Clarity Improvements (t2Ws, sXJP)**

* Clarified definition of source/target distributions
* Clarified that our use of “interpretability” refers to controllable intervention strength
* Switched from “ablation” to “sensitivity analysis” where appropriate
* Better explanation of why, in this low $n$ high $d$ regime, using the Sliced W distance is preferable.

**Technical Clarifications (QUnf, sXJP)**

* Clarified ReFT would require paired data for toxicity tasks, which we do not have access to.
* Explained hyperparameter choices ($λ_1=1, λ_G=1$, tune only γ)

Respectfully,

the authors

---

### Decision · Program_Chairs · 2025-09-17

**Decision:**

Accept (poster)

**Comment:**

The paper proposes LinEAS a distributional losses between activation of source and target distribution of activations.  The source activations are transformed via affine transport maps throughout all layer of the network , with a group sparsity regularization of the sliced Wasserstein loss between the distribution of the activations. Using this distributional loss allows using unpaired datasets versus previous work such as   ReFT this was clarified by author during rebuttal.  Experiments show that LinEAS effectiveness in  toxic reduction in LLMs and style control in text-image diffusion models.

Overall reviewers were positive of the papers and made suggestions to improve the paper in term of clarification of how the method compare to other baselines conceptually in terms of computational time. Other experimental suggestion in terms of a human study for their preference in terms of outputs of the method  as well as experiments on  effect of the regularization on the text to image experiments  , and a regularization path to show the pareto front as regularization parameter changes.

Some of these concerns and suggestions were addressed in the rebuttal, All these experiments, plots and clarifications should be added to the final manuscript.

Minor: the begining of the  abstract on openreview is cut off, please fix that.